# AROID: Improving Adversarial Robustness through Online Instance-wise Data Augmentation

## Abstract

Deep neural networks are vulnerable to adversarial examples. Adversarial training (AT) is an effective defense against adversarial examples. However, AT is prone to overfitting which degrades robustness substantially. Recently, data augmentation (DA) was shown to be effective in mitigating robust overfitting if appropriately designed and optimized for AT. This work proposes a new method to automatically learn online, instance-wise, DA policies to improve robust generalization for AT. This is the first automated DA method specific for robustness. A novel policy learning objective, consisting of Vulnerability, Affinity and Diversity, is proposed and shown to be sufficiently effective and efficient to be practical for automatic DA generation during AT. Importantly, our method dramatically reduces the cost of policy search from the 5000 hours of AutoAugment and the 412 hours of IDBH to 9 hours, making automated DA more practical to use for adversarial robustness. This allows our method to efficiently explore a large search space for a more effective DA policy and evolve the policy as training progresses. Empirically, our method is shown to outperform all competitive DA methods across various model architectures (CNNs and ViTs) and datasets (CIFAR10/100, Imagenette, ImageNet, SVHN). Our DA policy reinforced vanilla AT to surpass several state-of-the-art AT methods regarding both accuracy and robustness. It can also be combined with those advanced AT methods to further boost robustness.

## 1 Introduction

Deep neural networks (DNNs) are well known to be vulnerable to infinitesimal yet highly malicious artificial perturbations in their input, i.e., adversarial examples (Szegedy et al., 2014). Thus far, adversarial training (AT) has been the most effective defense against adversarial attacks (Athalye et al., 2018). AT is typically formulated as a min-max optimization problem:

$$\arg \min_{\boldsymbol{\theta}} \mathbb{E}[\arg \max_{\boldsymbol{\delta}} \mathcal{L}(\boldsymbol{x} + \boldsymbol{\delta}; \boldsymbol{\theta})] \tag{1}$$

where the inner maximization searches for the perturbation $\boldsymbol{\delta}$ to maximize the loss, while the outer minimization searches for the model parameters $\boldsymbol{\theta}$ to minimize the loss on the perturbed examples.

One major issue of AT is that it is prone to overfitting (Rice et al., 2020; Wong et al., 2020). Unlike in standard training (ST), overfitting in AT, a.k.a. robust overfitting (Rice et al., 2020), significantly impairs adversarial robustness. Many efforts (Li & Spratling, 2023b; Wu et al., 2020; Dong et al., 2022) have been made to understand robust overfitting and mitigate its effect. One promising solution is data augmentation (DA), which is a common technique to prevent ST from overfitting. However, many studies (Rice et al., 2020; Wu et al., 2020; Gowal et al., 2021; Rebuffi et al., 2021) have revealed that advanced DA methods, originally proposed for ST, often fail to improve adversarial robustness.Therefore, DA was usually combined with other regularization techniques such as Stochastic Weight Averaging (SWA) (Rebuffi et al., 2021), Consistency regularization (Tack et al., 2022) and Separate Batch Normalization (Addepalli et al., 2022) to improve its effectiveness. However, recent work (Li & Spratling, 2023c) demonstrated that DA alone can significantly improve AT if it has strong diversity and well-balanced hardness. This suggests that ST and AT may require different DA strategies, especially in terms of hardness. It is thus necessary to design DA schemes dedicated to AT.

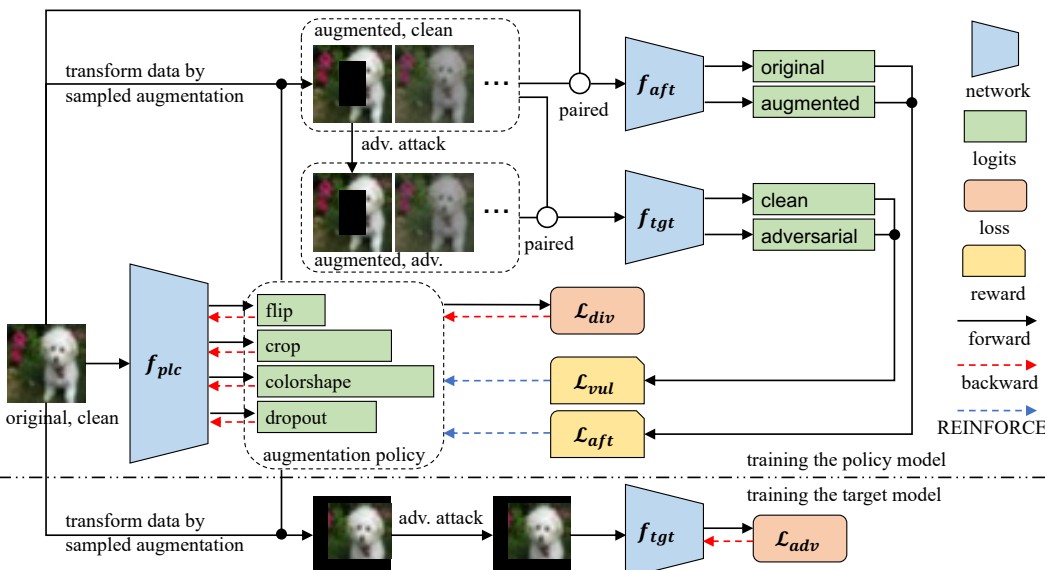

Figure 1: An overview of the proposed method (legend in the right column). The top part shows the pipeline for training the policy model, $f_{plc}$, while the bottom illustrates the pipeline for training the target model, $f_{tgt}$. $f_{aft}$ is a model pre-trained on clean data without any augmentation, which is used to measure the distribution shift caused by data augmentation. Please refer to Section 3 for a detailed explanation.

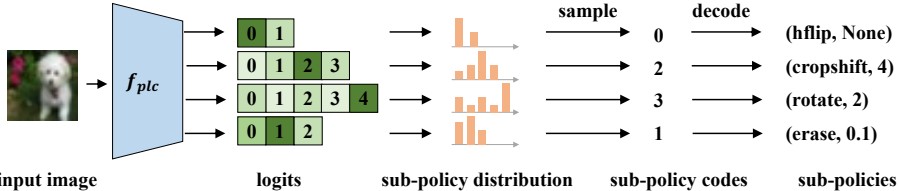

Figure 2: An example of the proposed augmentation sampling procedure. The policy model takes an image as input and outputs logit values defining multiple, multinomial, probability distributions corresponding to different sub-policies. A sub-policy code is created by sampling from each of these distributions, and decoded into a sub-policy, i.e., a transformation and its magnitude. These transformations are applied, in sequence, to augment the image.

IDBH (Li & Spratling, 2023c) is the first and the latest DA scheme specifically designed for AT. Despite its impressive robust performance, IDBH employs a heuristic search method to manually optimize DA. This search process requires a complete AT for every sampled policy, which induces prohibitive computational cost and scales poorly to large datasets and models. Hence, when the computational budget is limited, the hyperparameters for IDBH might be found using a reduced search space and by employing a smaller model, leading to compromised performance.

Another issue is that IDBH, in common with other conventional DA methods such as AutoAugment (Cubuk et al., 2019) and TrivialAugment (Müller & Hutter, 2021), applies the same strategy to all samples in the dataset throughout training. The distinctions between different training samples, and between the model checkpoints at different stages of training, are neglected. We hypothesize that different data samples at the same stage of training, as well as the same sample at the different stages of training, demand different DAs. Hence, we conjecture that an improvement in robustness could be realized by customizing DA for data samples and training stages.

To address the above issues, this work proposes a bi-level optimization framework (see Fig. 1) to automatically learn **A**dversarial **R**obustness by **O**nline **I**nstance-wise **D**ata-augmentation (AROID).

To the best of our knowledge, **AROID is the first automated DA method specific to adversarial robustness**. AROID employs a multi-head DNN-based policy model to map a data sample to a DA policy (see Fig. 2). This DA policy is defined as a sequence of pre-defined transformations applied with strength determined by the output of the policy model. This policy model is optimized, alongside the training of the target model, towards three novel objectives to achieve a target level of hardness and diversity. DA policies, therefore, are customized for each data instance and evolve with the target network as training progresses. This in practice produces a more globally optimal DA policy and thus benefits robustness. Importantly, the proposed policy learning objectives, in contrast to the conventional ones like validation accuracy (Cubuk et al., 2019), do not reserve a subset of the training data for validation and do not rely on prohibitively expensive inner loops for training the target model to evaluate the rewards of the sampled policies. The former ensures the entire training set is available for training to avoid potential data scarcity. The latter enables policy optimization to be much more efficient and scalable so that it is more practical for AT. Compared to IDBH in particular, this allows our approach to explore a larger space of DAs on the target. Taking an example of optimizing the DA for CIFAR10 and PRN18, AROID took 9 hours using an A100 GPU, IDBH took 412 hours using an A100 GPU, and AutoAugment took 5000 hours using a P100 GPU (Hataya et al., 2020).

Extensive experiments show that AROID outperforms all competitive DA methods across various datasets and model architectures while being more efficient than the prior art IDBH. **AROID achieves state-of-the-art robustness for DA methods** on the standard benchmarks. Besides, AROID outperforms, regarding accuracy and robustness, state-of-the-art AT methods. It also complements such robust training methods and can be combined with them to improve robustness further.

## 2 RELATED WORK

**Robust training**. To mitigate overfitting in AT, many methods other than DA, have been previously proposed. One line of works, IGR (Ross & Doshi-Velez, 2018), CURE (Moosavi-Dezfooli et al., 2019), AdvLC (Li & Spratling, 2023b), discovered a connection between adversarial vulnerability and the smoothness of input loss landscape, and promoted robustness by smoothing the input loss landscape. Meanwhile, Wu et al. (2020) and Chen et al. (2021) found that robust generalization can be improved by a flat weight loss landscape and proposed AWP and SWA, respectively, to smooth the weight loss landscape during AT. RWP (Yu et al., 2022) and SEAT (Wang & Wang, 2022) were later proposed to further refine AWP and SWA, respectively, to increase robustness. Many works, including MART (Wang et al., 2020), LAS-AT (Jia et al., 2022), ISEAT (Li & Spratling, 2023a), considered the difference between individual training instances and improved AT through regularizing in an instance-wise manner. Our proposed approach is also instance-wise, but contrary to existing methods tackles robust overfitting via DA instead of robust regularization. As shown in Section 4.2, it works well alone and, more importantly, complements the above techniques.

**Data augmentation for ST**. Although DA has been a common practice in many fields, we only review vision-based DA in this section as it is most related to our work. In computer vision, DA can be generally categorized as: basic, composite and mixup. Basic augmentations refer to a series of image transformations that can be applied independently. They mainly include crop-based (Random Crop (He et al., 2016a), Cropshift (Li & Spratling, 2023c), etc.), color-based (Brightness, Contrast, etc.), geometric-based (Rotation, Shear, etc.) and dropout-based (Cutout (DeVries & Taylor, 2017), Random Erasing (Zhong et al., 2020), etc.) transformations. Composite augmentations denote the composition of basic augmentations. Augmentations are composed into a single policy/schedule usually through two ways: interpolation (Hendrycks* et al., 2020; Wang et al., 2021) and sequencing (Cubuk et al., 2019; 2020; Müller & Hutter, 2021). Mixup (Zhang et al., 2018), and analogous works like Cutmix (Yun et al., 2019), can be considered as a special case of interpolation-based composition, which combines a pair of different images, instead of augmentations, as well as their labels to create a new image and its label.

Composite augmentations by design have many hyperparameters to optimize. Most previous works, as well as the pioneering AutoAugment (Cubuk et al., 2019), tackled this issue using automated machine learning (AutoML). DA policies were optimized towards maximizing validation accuracy (Cubuk et al., 2019; Lin et al., 2019; Li et al., 2020; Liu et al., 2021), maximizing training loss (Zhang et al., 2020) or matching the distribution density between the original and augmented data (Lim et al., 2019; Hataya et al., 2020). Optimization here is particularly challenging since DA operations are

usually non-differentiable. Major solutions seek to estimate the gradient of DA learning objective w.r.t. the policy generator or DA operations using, e.g., policy gradient methods (Cubuk et al., 2019; Zhang et al., 2020; Lin et al., 2019) or reparameterization trick (Li et al., 2020; Hataya et al., 2020). Alternative optimization techniques include Bayesian optimization (Lim et al., 2019) and population-based training (Ho et al., 2019). Noticeably, several works like RandAugment (Cubuk et al., 2020) and TrivialAugment (Müller & Hutter, 2021) found that if the augmentation space and schedule were appropriately designed, competitive results could be achieved using a simple hyperparameter grid search or fixed hyperparameters. This implies that in ST these advanced yet complicated methods may not be necessary. However, it remains an open question if simple search can still match these advanced optimization methods in AT. Besides, instance-wise DA strategy was also explored in Cheung & Yeung (2022); Miao et al. (2023) for ST. Our method is the first automated DA approach specific for AT. We follow the line of policy gradient methods to enable learning DA policies. A key distinction here is that our policy learning objective is designed to guide the learning of DA policies towards improved robustness for AT, while the objective of the above methods is to increase accuracy for ST.

## 3    OPTIMIZING DATA AUGMENTATION FOR ADVERSARIAL ROBUSTNESS

We propose a method to automatically learn DA alongside AT to improve robust generalization. An **instance-wise** DA policy is produced by a policy model and learned by optimizing the policy model towards three novel objectives. Updating of the policy model and the target model (the one being adversarially trained for the target task) alternates throughout training (the policy model is updated every $K$ updates of the target model), yielding an **online** DA strategy. This online, instance-adaptive, strategy produces different augmentations for different data instances at different stages of training.

The following notation is used. $x \in \mathbb{R}^d$ is a $d$-dimensional sample whose ground truth label is $y$. $x_i$ refers to $i$-th sample in a dataset. The model is parameterized by $\theta$. $\mathcal{L}(x, y; \theta)$ or $\mathcal{L}(x; \theta)$ for short denotes the predictive loss evaluated with $x$ w.r.t. the model $\theta$ (Cross-Entropy loss was used in all experiments). $\rho(x; \theta)$ computes the adversarial example of $x$ w.r.t. the model $\theta$. $p_i(x; \theta)$ refers to the output of the Softmax function applied to the final layer of the model, i.e., the probability at $i$-th logit given the input $x$.

### 3.1    MODELING THE DATA AUGMENTATION POLICY USING DNNS

Following the design of IDBH (Li & Spratling, 2023c) and TrivialAugment (Müller & Hutter, 2021), DA is implemented using four types of transformations: flip, crop, color/shape and dropout applied in order. We implement flip using HorizontalFlip, crop using Cropshift (Li & Spratling, 2023c), dropout using Erasing[1] (Zhong et al., 2020), and color/shape using a set of operations including Color, Sharpness, Brightness, Contrast, Autocontrast, Equalize, Shear (X and Y), Rotate, Translate (X and Y), Solarize and Posterize. A dummy operation, Identity, is included in each augmentation group to allow data to pass through unchanged. More details including the complete augmentation space are described in Appendix A.

To customize the DA applied to each data instance individually, a policy model parameterized by $\theta_{plc}$, is used to produce a DA policy conditioned on the input data (see Fig. 2). The policy model employs a DNN backbone to extract features from the data, and multiple, parallel, linear prediction heads on the top of the extracted features to predict the policy. The policy model used in this work has four heads corresponding to the four types of DA described above[2]. The output of a head is converted into a multinomial distribution where each logit represents a pre-defined sub-policy, i.e., an augmentation operation associated with a strength/magnitude (e.g. ShearX, 0.1). Different magnitudes of the same operation are represented by different logits, so that each has its own chance of being sampled. A particular sequence of sub-policies to apply to the input image are selected based on the probabilities encoded in the four heads of the policy network.

---

[1] Different from the original version applied at half chance, here erasing is always applied but the location and aspect ratio are randomly sampled from the given range.

[2] When training on SVHN only three heads were used, as HorizontalFlip is not appropriate for this dataset.

## 3.2 Objectives for Learning the Data Augmentation Policy

The policy model is trained using three novel objectives: (adversarial) Vulnerability, Affinity and Diversity. Vulnerability (Li & Spratling, 2023b) measures the loss variation caused by adversarial perturbation on the augmented data w.r.t. the target model:

$$\mathcal{L}_{vul}(\boldsymbol{x}; \boldsymbol{\theta}_{plc}) = \mathcal{L}(\rho(\hat{\boldsymbol{x}}; \boldsymbol{\theta}_{tgt}); \boldsymbol{\theta}_{tgt}) - \mathcal{L}(\hat{\boldsymbol{x}}; \boldsymbol{\theta}_{tgt}), \text{where } \hat{\boldsymbol{x}} = \Phi(\boldsymbol{x}; S(\boldsymbol{\theta}_{plc}(\boldsymbol{x}))) \tag{2}$$

$\Phi(\boldsymbol{x}; S(\boldsymbol{\theta}_{plc}(\boldsymbol{x})))$ augments $\boldsymbol{x}$ by $S(\boldsymbol{\theta}_{plc}(\boldsymbol{x}))$, the augmentations sampled from the output distribution of policy model conditioned on $\boldsymbol{x}$, so $\hat{\boldsymbol{x}}$ is the augmented data. A larger Vulnerability indicates that $\boldsymbol{x}$ becomes more vulnerable to adversarial attack after DA. A common belief about the relationship between training data and robustness is that AT benefits from adversarially hard samples. From a geometric perspective, maximizing Vulnerability encourages the policy model to project data into the previously less-robustified space. Nevertheless, the maximization of Vulnerability, if not constrained, would likely favor those augmentations producing samples far away from the original distribution. Training with such augmentations was observed to degrade accuracy and even robustness if accuracy overly reduced (Li & Spratling, 2023c). Therefore, Vulnerability should be maximized while the distribution shift caused by augmentation is constrained:

$$arg \max_{\boldsymbol{\theta}_{plc}} \mathcal{L}_{vul}(\boldsymbol{x}; \boldsymbol{\theta}_{plc}) \text{ s.t. } ds(\boldsymbol{x}, \hat{\boldsymbol{x}}) \leq D \tag{3}$$

where $ds(\cdot)$ measures the distribution shift between two samples and $D$ is a constant. Directly solving Eq. (3) is intractable, so we convert it into an unconstrained optimization problem by adding a penalty on the distribution shift as:

$$arg \max_{\boldsymbol{\theta}_{plc}} \mathcal{L}_{vul}(\boldsymbol{x}; \boldsymbol{\theta}_{plc}) - \lambda \cdot ds(\boldsymbol{x}, \hat{\boldsymbol{x}}) \tag{4}$$

where $\lambda$ is a hyperparameter and a larger $\lambda$ corresponds to a tighter constraint on distribution shift, i.e., smaller $D$. Distribution shift is measured using a variant of the Affinity metric (Gontijo-Lopes et al., 2021):

$$ds(\boldsymbol{x}, \hat{\boldsymbol{x}}) = \mathcal{L}_{aft}(\boldsymbol{x}; \boldsymbol{\theta}_{plc}) = \mathcal{L}(\hat{\boldsymbol{x}}; \boldsymbol{\theta}_{aft}) - \mathcal{L}(\boldsymbol{x}; \boldsymbol{\theta}_{aft}) \tag{5}$$

Affinity captures the loss variation caused by DA w.r.t. a model $\boldsymbol{\theta}_{aft}$ (called the affinity model): a model pre-trained on the original data (i.e., without any data augmentation). Affinity increases as the augmentation proposed by the policy network makes data harder for the affinity model to correctly classify. By substituting Eq. (5) into Eq. (4), we obtain an adjustable Hardness objective:

$$\mathcal{L}_{hrd}(\boldsymbol{x}; \boldsymbol{\theta}_{plc}) = \mathcal{L}_{vul}(\boldsymbol{x}; \boldsymbol{\theta}_{plc}) - \lambda \cdot \mathcal{L}_{aft}(\boldsymbol{x}; \boldsymbol{\theta}_{plc}) \tag{6}$$

This encourages the DA produced by the policy model to be at a level of hardness defined by $\lambda$ (larger values of $\lambda$ corresponding to lower hardness). Ideally, $\lambda$ should be tuned to ensure the distribution shift caused by DA is sufficient to benefit robustness while not being so severe as to harm accuracy.

Last, we introduce a Diversity objective to promote diverse DA. Diversity enforces a relaxed uniform distribution prior over the logits of the policy model, i.e., the output augmentation distribution:

$$\mathcal{L}_{div}^h(\boldsymbol{x}) = \frac{1}{C}[- \sum_{i}^{p_i^h < l} \log(p_i^h(\boldsymbol{x}; \boldsymbol{\theta}_{plc})) + \sum_{j}^{p_j^h > u} \log(p_j^h(\boldsymbol{x}; \boldsymbol{\theta}_{plc}))] \tag{7}$$

$C$ is the total count of logits violating either lower ($l$), or upper ($u$) limits and $h$ is the index of the prediction head. Intuitively speaking, the Diversity loss penalizes overly small and large probabilities, helping to constrain the distribution to lie in a pre-defined range $(l, u)$. As $l$ and $u$ approach the mean probability, the enforced prior becomes closer to a uniform distribution, which corresponds to a highly diverse DA policy. Diversity encourages the policy model to avoid the over-exploitation of certain augmentations and to explore other candidate augmentations. Note that Diversity is applied to the color/shape head in a hierarchical way: type-wise and strength-wise inside each type of augmentation.

Combining the above three objectives together, the policy model is trained to optimize:

$$arg \min_{\boldsymbol{\theta}_{plc}} -\mathbb{E}_{i \in B} \mathcal{L}_{hrd}(\boldsymbol{x}_i; \boldsymbol{\theta}_{plc}) + \beta \cdot \mathbb{E}_{h \in H} \mathcal{L}_{div}^h(\boldsymbol{x}; \boldsymbol{\theta}_{plc}) \tag{8}$$

where $B$ is the batch size and $\beta$ trades-off hardness against diversity. $\mathcal{L}_{div}^h$ is calculated across instances in a batch, so no need for averaging over $B$ like $\mathcal{L}_{hrd}$. The design of Eq. (8) reflects the prior that DA should have strong diversity and well-balanced hardness to be effective for AT (Li & Spratling, 2023c). Appendix B explains how the proposed method mitigates robust overfitting.

**Algorithm 1.** High-level training procedures of the proposed method. $\alpha$ is the learning rate. $M$ is the number of training iterations.

```
for i = 1 to M do
    // for every K iterations
    if i % K == 0 then
        // update the policy
            model by Algo. 2
    end
    // the policy distribution
    d = θ_plc(x_i)
    // sample & apply
        augmentations
    x̂_i = Φ(x_i; S(d))
    L = L(ρ(x̂_i; θ_tgt); θ_tgt)
    // update the target model
    θ_tgt = θ_tgt − α_tgt · ∇_{θ_tgt} L
end
```

**Algorithm 2.** Pseudo code of training the policy model for one iteration. $x$ is randomly sampled from the entire dataset.

```
d = θ_plc(x)
// same x used by all traj.
for t = 1 to T do
    x̂_(t) = Φ(x, S(d))
    P_(t) = ∏_{h=1}^H p_(t)^h  // prob of traj
                                    t
    L_hrd^(t)  // computed by Eq. (6)
end
L̃_hrd = (1/T) ∑_{t=1}^T L_hrd^(t)  // mean L_hrd^(t)
L = (1/T) ∑_{t=1}^T log(P_(t))[L_hrd^(t) − L̃_hrd]
L_div^(h)  // computed using Eq. (7)
L = −L + β (1/H) ∑_{h=1}^H L_div^(h)
θ_plc = θ_plc − α_plc · ∇_{θ_plc} L
```

## 3.3 OPTIMIZATION

The entire training is a bi-level optimization process (Algo. 1): the target and policy models are updated alternately. This online training strategy adapts the policy model to the varying demands for DA from the target model at the different stages of training. The target model is optimized using AT with the augmentation sampled from the policy model:

$$arg \min_{\boldsymbol{\theta}_{tgt}} \mathcal{L}(\rho(\Phi(\boldsymbol{x}; S(\boldsymbol{\theta}_{plc}(\boldsymbol{x}))); \boldsymbol{\theta}_{tgt}); \boldsymbol{\theta}_{tgt}) \tag{9}$$

After every $K$ updates of the target model, the policy model is updated using the gradients of the policy learning loss as follows:

$$\frac{Eq.\,(8)}{\partial \boldsymbol{\theta}_{plc}} = -\frac{\partial \mathbb{E}_{i\in B}\mathcal{L}_{hrd}(\boldsymbol{x}_i; \boldsymbol{\theta}_{plc})}{\partial \boldsymbol{\theta}_{plc}} + \beta \frac{\mathbb{E}_{h\in H}\mathcal{L}_{div}^h(\boldsymbol{x}; \boldsymbol{\theta}_{plc})}{\partial \boldsymbol{\theta}_{plc}} \tag{10}$$

The latter can be derived directly, while the former $\frac{\partial \mathcal{L}_{hrd}}{\partial \boldsymbol{\theta}_{plc}}$ cannot because the involved augmentation operations are non-differentiable. To estimate these gradients, we apply the REINFORCE algorithm (Williams, 1992) with baseline trick to reduce the variance of gradient estimation. It first samples $T$ augmentations, named trajectories, in parallel from the policy model and then computes the real Hardness value, $\mathcal{L}_{hrd}^{(t)}$, using Eq. (6) independently on each trajectory $t$. The gradients are estimated (see Appendix C for derivation) as follows:

$$\frac{\partial \mathbb{E}_{i\in B}\mathcal{L}_{hrd}(\boldsymbol{x}_i; \boldsymbol{\theta}_{plc})}{\partial \boldsymbol{\theta}_{plc}} \approx \frac{1}{B\cdot T}\sum_{i=1}^B \sum_{t=1}^T \sum_{h=1}^H \frac{\partial \log(p_{(t)}^h(\boldsymbol{x}_i; \boldsymbol{\theta}_{plc}))}{\partial \boldsymbol{\theta}_{plc}}[\mathcal{L}_{hrd}^{(t)}(\boldsymbol{x}_i; \boldsymbol{\theta}_{plc}) - \tilde{\mathcal{L}_{hrd}}] \tag{11}$$

$p_{(t)}^h$ is the probability of the sampled sub-policy at the $h$-th head and $\tilde{\mathcal{L}_{hrd}} = \frac{1}{T}\sum_{t=1}^T \mathcal{L}_{hrd}^{(t)}(\boldsymbol{x}_i; \boldsymbol{\theta}_{plc})$ is the mean $\mathcal{L}_{hrd}$ (the baseline used in the baseline trick) averaged over the trajectories. Algo. 2 illustrates one iteration of updating the policy model. Note that, when one model is being updated, backpropagation is blocked through the other. The affinity model, used in calculating the Affinity metric, is fixed throughout training. Appendix D discusses the stability of our method.

## 3.4 EFFICIENCY

The cost of AROID is composed of two parts: policy learning and DA sampling. Policy learning can be one-time expense if AROID is used in an offline way: DA policies are sampled from pre-trained policy models. DA sampling requires only one forward pass of the policy model, which can be negligible because the policy model can be much smaller than the target model while not hurting the performance. Therefore, AROID in offline mode is roughly as efficient as other regular DA methods.

Table 1: Comparison of the performance of various DA methods. The **best** and second best results are highlighted in each column. The baseline augmentation was Horizontal Flip plus Random Crop.

| DA Method | CIFAR10 | | | | CIFAR100 | | | | Imagenette | |
|---|---|---|---|---|---|---|---|---|---|---|
| | WRN34-10 | | ViT-B/4 | | WRN34-10 | | PRN18 | | ViT-B/16 | |
| | Acc. | Rob. | Acc. | Rob. | Acc. | Rob. | Acc. | Rob. | Acc. | Rob. |
| baseline | 85.83 | 52.26 | 83.04 | 46.72 | 61.44 | 27.98 | 55.04 | 24.83 | 92.73 | 66.47 |
| Cutout | 86.95 | 52.89 | 83.61 | 48.67 | 59.04 | 27.51 | 57.37 | 24.51 | 93.27 | 67.20 |
| Cutmix | 86.88 | 53.38 | 80.83 | 47.24 | 58.57 | 27.49 | 57.32 | 25.54 | 93.87 | 70.20 |
| AutoAugment | 87.71 | 54.60 | 81.96 | 47.47 | **64.10** | 29.08 | 58.51 | 25.28 | 95.13 | 67.60 |
| TrivialAugment | 87.35 | 53.86 | 80.55 | 46.39 | 62.55 | 28.97 | 57.24 | 24.82 | **95.25** | 69.00 |
| IDBH | 88.61 | 55.29 | 85.09 | 49.63 | 60.93 | 29.03 | 59.38 | 26.24 | 95.20 | 69.93 |
| AROID (ours) | **88.99** | **55.91** | **87.34** | **51.25** | 64.44 | **29.75** | **60.17** | **26.56** | 94.88 | **71.32** |

In online mode, in the worst case, AROID adds about 43.6% extra computation to baseline AT (see calculation in Appendix E.1) when $T = 8$ and $K = 5$. This is less than the overhead 52.5% of the state-of-the-art AT method LAS-AT (Jia et al., 2022) and substantially less than the search cost of IDBH and AutoAugment (compared in Section 4.5). Furthermore, we observed that AROID can still achieve robustness higher than other competitors with a much smaller policy model (Appendix G.5.2), reduced $T$ and increased $K$ (Section 4.5) for improved efficiency. For example, setting $T = 4$ and $K = 20$, the overhead is only about 10% compared to baseline AT. Another efficiency concern, as for all other deep learning methods, is hyperparameter optimization. Appendix E.2 discusses how this can be done efficiently so that AROID can be fast adapted to a new setting.

## 4 EXPERIMENTS

The experiments in this section were based on the following setup unless otherwise specified. We used model architectures WideResNet34-10 (WRN34-10) (Zagoruyko & Komodakis, 2016), Vision Transformer (ViT-B/16 and ViT-B/4) (Dosovitskiy et al., 2021) and PreAct ResNet-18 (PRN18) (He et al., 2016b). We used $\ell_\infty$ PGD10 for AT and AutoAttack (Croce & Hein, 2020) for evaluating adversarial robustness. By default, AROID is trained with $T = 8$ and $K = 5$. Please refer to Appendix F for the detailed experimental settings and the values of other hyper-parameters.

### 4.1 BENCHMARKING DATA AUGMENTATION ON ADVERSARIAL ROBUSTNESS

Tab. 1 compares our proposed method against existing DA methods. **AROID outperforms all existing methods regarding robustness across all four tested settings**. The improvement over the previous best method is particularly significant for ViT-B on CIFAR10 (+1.62%) and Imagenette (+1.12%). Note that in most cases IDBH is the only method whose robustness is close to ours. However, our method is much more efficient than IDBH in terms of policy search (shown in Section 4.5). If our method is compared only to those methods with a computational cost the same or less than AROID's, i.e., excluding IDBH and AutoAugment, the improvement over the second best method is +2.05%/2.58%/1.12%/1.02% for the four experiments. Furthermore, we highlight the substantial improvement over the baseline of our method, +3.65%/4.53%/4.85%/1.73%, in these four settings.

In addition, **AROID also achieves the highest accuracy in three of the four tested settings**, and in the fourth setting (Imagenette) the accuracy gap between the best method and ours is marginal (0.37%). Overall, our method significantly improves both accuracy and robustness, achieving a much better trade-off between accuracy and robustness. The consistent superior performance of our method, across various datasets (low and high resolution, simple and complex) and model architectures (CNNs and ViTs, small and large capacity), suggests that it has a good generalization ability. To ensure the reliability of our evaluation, the result of robustness evaluated by alternative attacks is given in Appendix G.1.

Table 2: The performance of various robust training (RT) methods with baseline (HorizontalFlip+RandomCrop) and our augmentations for WRN34-10 on CIFAR10.

| RT method | DA method | Accuracy | Robustness |
|---|---|---|---|
| AT (Madry et al., 2018) | baseline | 85.83±.76 | 52.26±.02 |
| AT-SWA (Rebuffi et al., 2021) | baseline | 84.30±.14 | 54.29±.15 |
| AT-AWP (Wu et al., 2020) | baseline | 85.93±.25 | 54.34±.40 |
| AT-RWP (Yu et al., 2022) | baseline | 86.86 ±.51 | 54.61±.11 |
| MART (Wang et al., 2020) | baseline | 84.17 | 51.10 |
| MART-AWP (Wu et al., 2020) | baseline | 84.43 | 54.23 |
| SEAT (Wang & Wang, 2022) | baseline | 86.44±.12 | 55.67±.22 |
| LAS-AT (Jia et al., 2022) | baseline | 86.23 | 53.58 |
| LAS-AWP (Jia et al., 2022) | baseline | 87.74 | 55.52 |
| AT (Madry et al., 2018) | AROID (ours) | **88.99**±.24 | **55.91**±.25 |
| AT-SWA | AROID (ours) | 87.84±.16 | 56.67±.21 |
| AT-AWP | AROID (ours) | 87.94±.11 | 56.98±.20 |
| AT-AWP-SWA | AROID (ours) | **88.39**±.10 | **57.03**±.01 |

Table 3: The result of AROID on ImageNet with ConvNeXt-T.

| DA method | Accuracy | Robustness |
|---|---|---|
| baseline | 71.22 | 36.22 |
| AutoAugment | 70.42 | 37.80 |
| AROID (ours) | **71.62** | **40.40** |

Table 4: The performance of our method when the policy model is pre-trained (AROID-T) or trained on-the-fly (AROID) for WRN34-10 on CIFAR10.

| Policy source | Accuracy | Robustness |
|---|---|---|
| AROID-T | 88.76 ± .14 | 55.61 ± .14 |
| AROID | **88.99** ± .24 | **55.91** ± .25 |

## 4.2 COMPARISON WITH STATE-OF-THE-ART ROBUST TRAINING METHODS

Tab. 2 compares our method against state-of-the-art robust training methods. It can be seen that AROID substantially improves vanilla AT in terms of accuracy (by 3.16%) and robustness (by 3.65%). This improvement is sufficient to boost the performance of vanilla AT to surpass the state-of-the-art robust training methods like SEAT and LAS-AWP in terms of both accuracy and robustness. This suggests that our method achieved a better trade-off between accuracy and robustness while boosting robustness. More importantly, our method, as it is based on DA, can be easily integrated into the pipeline of existing robust training methods and, as our results show, is complementary to them. By combining with SWA and/or AWP, our method substantially improves robustness even further while still maintaining an accuracy higher than that achieved by others methods. Appendix G.3 compares AROID against more state-of-the-art methods.

## 4.3 GENERALIZATION TO A LARGE-SCALE DATASET

To further test the generalizability and scalability of our method to a large-scale dataset, we train AROID on ImageNet (Deng et al., 2009) with ConvNeXt-T (Liu et al., 2022). Some DA methods are missing in this comparison due to the limit of computational resource (explained in Appendix F.4). As shown in Tab. 3, AROID significantly improves robustness over the baseline by 4.18% and AutoAugment by 2.6%. It also achieves the highest accuracy among the tested methods. Overall, AROID is able to scale and generalize to ImageNet.

## 4.4 TRANSFERABILITY OF A LEARNED DATA AUGMENTATION POLICY

This section assesses the effectiveness of offline AROID: sampling DA policies from pre-trained policy models. This transferred version of AROID is called AROID-T. At each epoch of training the target network, AROID-T uses a policy network checkpoint saved at the corresponding epoch when using AROID. We consider the case where AROID-T is used to train a target network with the same architecture on the same dataset as was used when creating the policy network checkpoints. We did not test the transferability across different training setups such as model architectures and robust training methods, because it is expected that the required DA policies will differ, especially when the capacity of the target model is considerably different (Li & Spratling, 2023c).

As shown in Tab. 4, AROID-T achieved a robustness of 55.61% which is slightly lower than that of AROID (55.91%). Note that the result of AROID-T is still better than that of the previous best DA-based method (IDBH 55.29%, see Tab. 1), and is close to the result of the best robust training method (SEAT 55.67%, see Tab. 2). Therefore, the policy learned by AROID is able to transfer to a reasonable degree, at least when using the same architecture and training settings.

Table 5: The cost of policy search for automated DA methods using PRN18 on CIFAR10. AROID is used in online mode. The size of search space counts the possible combinations of probabilities and magnitudes. Our search space is uncountable due to its continuous range of probability, and is much larger than that of IDBH as it covers a much wider range of probabilities and magnitudes. Time denotes the total hours required for one search over the search space using an Nvidia A100 GPU for IDBH and AROID and a P100 GPU for AutoAugment (data is copied from Hataya et al. (2020)).

| Method | K | T | Acc. | Rob. | Search Space | | | | Time |
|--------|---|---|------|------|-----------|-------------|-----------|------|------|
| | | | | | Prior dep. | Probability | Magnitude | Size | |
| AutoAugment | - | - | 83.27 | 49.20 | No | discrete | discrete | $2.9 \times 10^{32}$ | 5000 |
| IDBH | - | - | 84.23 | 50.47 | Yes | discrete | discrete | 80 | 412.83 |
| AROID | 5 | 8 | **84.68** | **50.57** | No | continuous | discrete | uncountable | 9.51 |
| AROID | 20 | 8 | 84.11 | 50.45 | No | continuous | discrete | uncountable | 6.85 |
| AROID | 20 | 4 | 83.63 | 50.52 | No | continuous | discrete | uncountable | **6.24** |

### 4.5 COMPARISON OF POLICY SEARCH COSTS

We compare here the cost of policy search of AROID against other automated DA methods, i.e., AutoAugment and IDBH. Before comparison, it is important to be aware that the search cost for IDBH increases linearly with the size of search space, while the cost of AROID stays approximately constant. IDBH thus uses a reduced search space that is much smaller than the search space of AROID. However, reducing the search space depends on prior knowledge about the training datasets, which may not generalize to other datasets. Moreover, scaling IDBH to our larger search space is intractable, and it would be even more intractable if IDBH was applied to find DAs for each data instance at each stage of training, as is done by AROID.

Even in the most expensive configuration ($K = 5$ and $T = 8$), AROID is substantially cheaper than IDBH and AutoAugment regarding the cost of policy search as shown in Tab. 5. The computational efficiency of AROID can be further increased by reducing the policy update frequency (increasing $K$) and/or decreasing the number of trajectories $T$, while still matching the robustness of IDBH. If IDBH and AutoAugment were restricted to use the same, much lower, budget for searching for a DA policy, given the huge gap, we suspect that they may find nothing useful. Last, even ignoring the training time of the DA policy and comparing AROID used in an offline manner, AROID-T, it still outperforms IDBH and AutoAugment (Tabs. 1 and 4).

### 4.6 ABLATION STUDY AND VISUALIZATION

Ablation study is conducted in Appendix G.5 including the sensitivity of AROID to its hyperparameters and the architecture of policy model, and the comparison of learned policy to uniform sampling. Besides, the learned DA policies are visualized in Appendix H. We observe that the learned DA policies vary among instances and evolve during training. Furthermore, the augmentation preference of the learned DA policies is consistent to the previous findings (Cubuk et al., 2019; Rebuffi et al., 2021), which verifies AROID's effectiveness.

## 5 CONCLUSIONS

This work introduces an approach, dubbed AROID, to efficiently learn online, instance-wise, DA policies for improved robust generalization in AT. AROID is the first automated DA method specific for AT. Extensive experiments show its superiority over both alternative DA methods and contemporary AT methods in terms of accuracy and robustness. This confirms the necessity of optimizing DA for improved adversarial robustness. The learned DA policies are visualized to verify the effectiveness of AROID and understand the preference of AT for DA.

However, AROID has some limitations as well. First, despite being more efficient than IDBH, it still adds extra computational burden to training, unless AROID-T is used. This could harm its scalability to larger datasets and model architectures. Second, the Diversity objective enforces a minimal chance (set by the lower limit) of applying harmful transformations and/or harmful magnitudes if they are

included in the search space. This constrains the ability of AROID to explore a wider (less filtered) search space. Future works could investigate more efficient AutoML algorithms for learning DA policies for AT, and design new policy learning objectives to reduce the number of hyperparameters and alleviate the side-effect of Diversity.

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

Table 6: Augmentation space

| Flip | | | Crop | | | Color/Shape | | | Dropout | | |
|---|---|---|---|---|---|---|---|---|---|---|---|
| operations | magnitudes | count | operations | magnitudes | count | operations | magnitudes | count | operations | magnitudes | count |
| Identity | - | 1 | Identity | - | 1 | Identity | - | 1 | Identity | - | 1 |
| Horiz. Flip | - | 1 | Cropshift | 1, 2, 3, 4, 5, 6, 7, 8, 9, 10, 11, 12, 13, 14, 15 | 15 | Autocontrast | - | 1 | Erasing | .05, .10, .15, .20, .25, .30, .35, .40, .45, .50 | 10 |
| | | | | | | Equalize | - | 1 | | | |
| | | | | | | Posterize | 4, 5, 6, 7, 8 | 5 | | | |
| | | | | | | Solarize | 25, 51, 76, 102, 128, 153, 179, 204, 230, 256 | 10 | | | |
| | | | | | | Rotate | 3, 6, 9, 12, 15, 18, 21, 24, 27, 30 | 10 | | | |
| | | | | | | ShearX | .03, .06, .09, .12, .15, .18, .21, .24, .27, .30 | 10 | | | |
| | | | | | | ShearY | .03, .06, .09, .12, .15, .18, .21, .24, .27, .30 | 10 | | | |
| | | | | | | TranslateX | 1, 2, 3, 4, 5, 6, 7, 8, 9, 10 | 10 | | | |
| | | | | | | TranslateY | 1, 2, 3, 4, 5, 6, 7, 8, 9, 10 | 10 | | | |
| | | | | | | Color | .28, .46, .64, .82, 1.0, 1.18, 1.36, 1.54, 1.72, 1.9 | 10 | | | |
| | | | | | | Contrast | .28, .46, .64, .82, 1.0, 1.18, 1.36, 1.54, 1.72, 1.9 | 10 | | | |
| | | | | | | Brightness | .28, .46, .64, .82, 1.0, 1.18, 1.36, 1.54, 1.72, 1.9 | 10 | | | |
| | | | | | | Sharpness | .28, .46, .64, .82, 1.0, 1.18, 1.36, 1.54, 1.72, 1.9 | 10 | | | |

# A  DA SEARCH SPACE FOR AROID

Tab. 6 shows the complete DA search space used by AROID. For Color/Shape group, we adopted the same operations as RandAugment's, but discretize the range of magnitudes for each operation into 10 even values if possible. For Erasing in Dropout group, the magnitude corresponds to the scale (the proportion of erased area against input image), while the aspect ratio (of erased area) is uniformly sampled from range $(0.3, 3.3)$. The search space only defines the operations and their magnitudes, while the probabilities of applying these operations are learned by AROID.

# B  AROID'S MECHANISM FOR MITIGATING ROBUST OVERFITTING

Below is some insight into the effectiveness of AROID. Robust generalization has been shown in Schmidt et al. (2018) to require much more data than ST. That's why data augmentation in general can alleviate robust overfitting. Next, focusing on "dynamic", it has been observed in ST that different training stages (Hataya et al., 2022; Lin et al., 2019) and different classes/instances (Cheung & Yeung, 2022) prefer different data augmentations. Regarding AT, a relevant observation is that robust overfitting occurs with the degradation of training adversary throughout training (Li & Spratling, 2023b). Therefore, dynamic DA mitigates robust overfitting by dynamically tuning DA policy to keep appropriately hard or even become progressively harder over the course of AT. For example, imaging that some data augmentations are adversarially overfitted by the underlying model (i.e. adversarial examples generated on the augmented data become easier to correctly classify) at the early stages of AT, dynamic DA counters this overfitting by reducing their probability to be sampled and raising the chance of sampling other potential underfitted yet beneficial data augmentations.

Specific to the objectives of policy learning, the Vulnerability objective is calculated based on the feedback of adversarial vulnerability from the target model. Therefore, the policy model learns from the feedback of target model about what type and strength of DA increases adversarial vulnerability. The Affinity objective is then used to constrain data augmentation to be not so overly hard that it impairs the performance. The Diversity objective prevents the over-exploitation of particular data augmentations and encourages the exploration of a diverse range of data augmentations. These three objectives jointly determine what DA are necessary for a training sample.

## C   DETAILED DERIVATION

This section discusses how we derive the gradients of Hardness metric w.r.t. the parameters of the policy model:

$$\frac{\partial \mathbb{E}_{i \in B} \mathcal{L}_{hrd}(\boldsymbol{x}_i; \boldsymbol{\theta}_{plc})}{\partial \boldsymbol{\theta}_{plc}} \tag{12}$$

First, we rewrite Eq. (12) as below, so that we can focus on the gradient derivation part.

$$\frac{1}{B} \sum_{i=1}^{B} \frac{\partial \mathcal{L}_{hrd}(\boldsymbol{x}_i; \boldsymbol{\theta}_{plc})}{\partial \boldsymbol{\theta}_{plc}} \tag{13}$$

Next, to apply the REINFORCE algorithm, we substitute the gradient of the $\mathcal{L}_{hrd}$ for a sampled trajectory in Eq. (13) with the gradient of the expected $\mathcal{L}_{hrd}$ for multiple sampled trajectories as

$$\frac{1}{B} \sum_{i=1}^{B} \frac{\partial \mathbb{E}_{t \in T} \mathcal{L}_{hrd}^{(t)}(\boldsymbol{x}_i; \boldsymbol{\theta}_{plc})}{\partial \boldsymbol{\theta}_{plc}} \tag{14}$$

By applying the REINFORCE algorithm, we have (batch averaging is omitted for simplicity)

$$\frac{\partial \mathbb{E}_{t \in T} \mathcal{L}_{hrd}^{(t)}(\boldsymbol{x}_i; \boldsymbol{\theta}_{plc})}{\partial \boldsymbol{\theta}_{plc}} = \frac{\partial \sum_{t=1}^{T} \mathcal{P}_{(t)}(\boldsymbol{x}_i; \boldsymbol{\theta}_{plc}) \mathcal{L}_{hrd}^{(t)}(\boldsymbol{x}_i; \boldsymbol{\theta}_{plc})}{\partial \boldsymbol{\theta}_{plc}} \tag{15}$$

$$= \sum_{t=1}^{T} \frac{\partial \mathcal{P}_{(t)}(\boldsymbol{x}_i; \boldsymbol{\theta}_{plc})}{\partial \boldsymbol{\theta}_{plc}} \mathcal{L}_{hrd}^{(t)}(\boldsymbol{x}_i; \boldsymbol{\theta}_{plc}) \tag{16}$$

$$= \sum_{t=1}^{T} \mathcal{P}_{(t)}(\boldsymbol{x}_i; \boldsymbol{\theta}_{plc}) \frac{\partial \log(\mathcal{P}_{(t)}(\boldsymbol{x}_i; \boldsymbol{\theta}_{plc}))}{\partial \boldsymbol{\theta}_{plc}} \mathcal{L}_{hrd}^{(t)}(\boldsymbol{x}_i; \boldsymbol{\theta}_{plc}) \tag{17}$$

$$= \mathbb{E}_{i \in T} \frac{\partial \log(\mathcal{P}_{(t)}(\boldsymbol{x}_i; \boldsymbol{\theta}_{plc}))}{\partial \boldsymbol{\theta}_{plc}} \mathcal{L}_{hrd}^{(t)}(\boldsymbol{x}_i; \boldsymbol{\theta}_{plc}) \tag{18}$$

$\mathcal{P}_{(t)}(\boldsymbol{x}_i; \boldsymbol{\theta}_{plc})$ is the probability of sampled trajectory. Following the previous practices (Zhang et al., 2020; Lin et al., 2019; Jia et al., 2022), we approximate Eq. (18) as

$$\frac{\partial \mathbb{E}_{t \in T} \mathcal{L}_{hrd}^{(t)}(\boldsymbol{x}_i; \boldsymbol{\theta}_{plc})}{\partial \boldsymbol{\theta}_{plc}} \approx \frac{1}{T} \sum_{t=1}^{T} \frac{\partial \log(\mathcal{P}_{(t)}(\boldsymbol{x}_i; \boldsymbol{\theta}_{plc}))}{\partial \boldsymbol{\theta}_{plc}} \mathcal{L}_{hrd}^{(t)}(\boldsymbol{x}_i; \boldsymbol{\theta}_{plc}) \tag{19}$$

Next, by expanding $\mathcal{P}_{(t)} = \prod_{h=1}^{H} p_{(t)}^{h}$, we have

$$\frac{\partial \mathbb{E}_{t \in T} \mathcal{L}_{hrd}^{(t)}(\boldsymbol{x}_i; \boldsymbol{\theta}_{plc})}{\partial \boldsymbol{\theta}_{plc}} \approx \frac{1}{T} \sum_{t=1}^{T} \frac{\partial \log(\prod_{h=1}^{H} p_{(t)}^{h}(\boldsymbol{x}_i; \boldsymbol{\theta}_{plc}))}{\partial \boldsymbol{\theta}_{plc}} \mathcal{L}_{hrd}^{(t)}(\boldsymbol{x}_i; \boldsymbol{\theta}_{plc}) \tag{20}$$

$$\approx \frac{1}{T} \sum_{t=1}^{T} \frac{\partial \sum_{h=1}^{H} \log(p_{(t)}^{h}(\boldsymbol{x}_i; \boldsymbol{\theta}_{plc}))}{\partial \boldsymbol{\theta}_{plc}} \mathcal{L}_{hrd}^{(t)}(\boldsymbol{x}_i; \boldsymbol{\theta}_{plc}) \tag{21}$$

$$\approx \frac{1}{T} \sum_{t=1}^{T} \sum_{h=1}^{H} \frac{\partial \log(p_{(t)}^{h}(\boldsymbol{x}_i; \boldsymbol{\theta}_{plc}))}{\partial \boldsymbol{\theta}_{plc}} \mathcal{L}_{hrd}^{(t)}(\boldsymbol{x}_i; \boldsymbol{\theta}_{plc}) \tag{22}$$

To reduce the variance of gradient estimation, we apply the baseline trick by subtracting mean value, $\tilde{\mathcal{L}_{hrd}} = \frac{1}{T} \sum_{t=1}^{T} \mathcal{L}_{hrd}^{(t)}(\boldsymbol{x}_i; \boldsymbol{\theta}_{plc})$, from $\mathcal{L}_{hrd}^{(t)}$ as

$$\frac{\partial \mathbb{E}_{t \in T} \mathcal{L}_{hrd}^{(t)}(\boldsymbol{x}_i; \boldsymbol{\theta}_{plc})}{\partial \boldsymbol{\theta}_{plc}} \approx \frac{1}{T} \sum_{t=1}^{T} \sum_{h=1}^{H} \frac{\partial \log(p_{(t)}^{h}(\boldsymbol{x}_i; \boldsymbol{\theta}_{plc}))}{\partial \boldsymbol{\theta}_{plc}} [\mathcal{L}_{hrd}^{(t)}(\boldsymbol{x}_i; \boldsymbol{\theta}_{plc}) - \tilde{\mathcal{L}_{hrd}}] \tag{23}$$

Eventually, by adding back the batch averaging, we have our ultimate form of gradients as

$$\frac{\partial \mathbb{E}_{i \in B} \mathbb{E}_{t \in T} \mathcal{L}_{hrd}^{(t)}(\boldsymbol{x}_i; \boldsymbol{\theta}_{plc})}{\partial \boldsymbol{\theta}_{plc}} \approx \frac{1}{B \cdot T} \sum_{i=1}^{B} \sum_{t=1}^{T} \sum_{h=1}^{H} \frac{\partial \log(p_{(t)}^{h}(\boldsymbol{x}_i; \boldsymbol{\theta}_{plc}))}{\partial \boldsymbol{\theta}_{plc}} [\mathcal{L}_{hrd}^{(t)}(\boldsymbol{x}_i; \boldsymbol{\theta}_{plc}) - \tilde{\mathcal{L}_{hrd}}]$$

$$\tag{24}$$

## D  STABILITY OF ALTERNATED TRAINING OF AROID

We have tested AROID on 5 datasets (CIFAR10, CIFAR100, SVHN, Imagenette, ImageNet), 5 adversarial training methods (PGD, SCORE, TRADES, AWP, SWA) and 4 model architectures (PRN, WRN, ViT, ConvNeXt with varied model size) with numerous ablation studies of 6 hyper-parameters, $\lambda$, $\beta$, diversity limits, $T$ and $K$. We have not observed any stability issue in all these experiments. Besides, the variance of performance across multiple runs in the same setting is small (Tab. 1) further confirming the stability and reproducibility of AROID. Furthermore, a similar alternating process of training a policy model using the REINFORCE algorithm has been adopted in several previous works like LAS-AT (Jia et al., 2022), OHL (Lin et al., 2019) etc., and none of them reported any stability issue. We see no reason for our method to be an exception. Overall, the experience from previous works and our experimental data both suggest that AROID has no issue of stability/reproducibility.

## E  EFFICIENCY ANALYSIS

### E.1  POLICY LEARNING

The efficiency of AROID is analyzed here. $F_t/F_p/F_a$ and $B_t/B_p/B_a$ denote the cost of forward and backward pass on target/policy/affinity model respectively. For each iteration of updating policy model, the major overhead is

- Predict DA distribution: 1 $F_p$
- Vulnerability: for each of $T$ trajectories, 2 $(F_t + B_t)$ to generate adversarial examples and 1 $F_t$ to calculate loss. Overall, $(3F_t + 2B_t)T$
- Affinity: 1 $F_a$ to calculate the loss of original data which is shared by all $T$ trajectories. 1 $F_a$ to calculate the loss of augmented data for each of $T$ trajectories. Overall, $(F_aT + F_a)$
- Diversity: the calculation of diversity loss adds negligible overhead and does not require $F$ or $B$
- Update policy model: 1 $B_p$

To sum up, one iteration of policy update costs

$$(3F_t + 2B_t)T + (F_aT + F_a) + F_p + B_p \tag{25}$$

Policy model is updated every $K$ iterations of target model, so the averaged policy learning cost per iteration of target model training is

$$[(3F_t + 2B_t)T + (F_aT + F_a) + F_p + B_p]/K \tag{26}$$

The overall overhead of AROID is learning cost plus 1 $F_p$ for every iteration of target model to sample DA, so

$$[(3F_t + 2B_t)T + (F_aT + F_a) + F_p + B_p]/K + F_p \tag{27}$$

In worst case, policy and affinity models use the same architecture as target model, so the cost is

$$[(4T + 2)/K + 1]F_t + (2T + 1)B_t/K \tag{28}$$

The most expensive setting we use is $T = 8$ and $K = 5$, so it costs $7.8F_t + 3.4B_t$ roughly, assuming $2F_t = 1B_t$, $4.8(F_t + B_t)$ in addition to $11(F_t + B_t)$ of underlying PGD10 AT. Overall, in worst case, AROID adds about 43.6% extra computation to baseline AT. For a cheaper setting $T = 4$ and $K = 20$, the overhead is roughly $1.9F_t + 0.45B_t$ about 10% more than baseline AT.

### E.2  HYPERPARAMETER OPTIMIZATION

First, as shown in Appendix G.5.1, most of our hyperparameters can transfer well among different training settings, so that only a light tuning is needed to achieve reasonably good performance for new setting. In most cases, only $\lambda$ needs to be tuned. Second, hyperparameter optimization can be accelerated by first searching with a cheap setting like $K = 20$ and $T = 4$ and then transferring the found values to the final setting, i.e., $K = 5$ and $T = 8$. Note that our hyperparameter tuning process is not different from others. Some related works like LAS-AT also have multiple hyper-parameters to tune.

# F    EXPERIMENTAL SET-UPS

## F.1    GENERAL TRAINING SETTINGS

For CIFAR10/100, models were trained by stochastic gradient descent (SGD) for 200 epochs with an initial learning rate 0.1 divided by 10 at 50% and 75% of epochs. The momentum was 0.9, the weight decay was 5e-4 and the batch size was 128.

The experiments on Imagenette [3] and SVHN followed a similar protocol as those on CIFAR10 except the following changes. The initial learning rate on SVHN was 0.01. For Imagenette, the weight decay was 1e-4, the total number of epochs was 40, and the learning rate was decayed at 36th and 38th epoch. The ViT-B/16 was pre-trained on ImageNet-1K (Deng et al., 2009). Gradient clipping was applied throughout training. Note that CIFAR10 with ViT-B/4 is trained using the same setting as Imagenette with ViT-B/16.

For ImageNet, models were trained for 50 epochs with an initial learning rate 0.01 divided by 10 at 20th and 40th epoch. Models were pre-trained on ImageNet-1K. The weight decay was 0.

Experiments were run on Tesla V100 and A100. All results reported by us were averaged over 3 runs except for ImageNet due to the limit of computational resource.

## F.2    ADVERSARIAL TRAINING SETTINGS

By default, we used $\ell_\infty$ projected gradient descent (Madry et al., 2018) with a perturbation budget, $\epsilon$, of 8/255. The number of steps was 10 and the step size was 2/255 for CIFAR10 and 1/255 for SVHN. To stabilize the training on SVHN, the perturbation budget, $\epsilon$, was increased from 0 to $\epsilon$ linearly in the first five epochs and then kept constant for the remaining epochs, as suggested by Andriushchenko & Flammarion (2020). Note that, following Rice et al. (2020), we tracked PGD10 robustness on the test set at the end of each epoch during training and selected the checkpoint with the highest PGD10 robustness, i.e., the "best" checkpoint to report robustness.

For ImageNet, the perturbation budget, $\epsilon$, was 4/255, the number of steps was 2 and the step size was $2\epsilon/3$. The same warm-up strategy as used in SVHN was adopted.

## F.3    CONFIGURATION OF AROID

Vulnerability objective was calculated based on PGD2 with a step size of 2/255 except that PGD1 with a step size of 4/255 for ImageNet. The affinity models used the same architecture as the target model. The affinity models were pre-trained using ST with the same settings as their AT trained counterparts yet with no augmentation. Early stopping was used if training accuracy was close to 100%. The policy model's backbone was PRN18 on CIFAR10 and SVHN, and ViT-B/16 (pre-trained on ImageNet-1K) on Imagenette as it was observed to be difficult for PRN18 to quickly fit Imagenette data to a reasonable degree in ST. Note that this ability is especially important when training on Imagenette because the total number of epochs (40) is much less than for the other datasets (200). The policy model was trained using SGD with a constant learning rate (0.001 for CIFAR10 and SVHN and 0.1 for Imagenette due to the reduced number of training epochs) and the same momentum as the target optimizer's. Gradient clipping was applied to stabilize the training of the policy model. In the initial five epochs of training, we did not train the policy model nor apply it to augment the data (no augmentation at all was applied) since the target model changed rapidly. When training on CIFAR10, we progressively hardened DA by decreasing $\lambda$ as the learning rate decayed since this improved robustness over the constant $\lambda$ scheme. By default, $T = 5$ and $K = 8$ are used except that for ImageNet $T = 20$ and $K = 4$ due to the limit of computational resource. The value of main hyperparameters used in our experiments are summarized in Tab. 7.

---

[3]Imagenette is a subset of ImageNet (Deng et al., 2009) consisting of 10 classes. We adopt a previous version (v1), `https://s3.amazonaws.com/fast-ai-imageclas/imagenette.tgz`, as suggested by (Mo et al., 2022).

Table 7: The value of hyperparameters of our method used in various training settings. The value of $l$ and $u$ listed here is a factor relative to the arithmetic mean chance, $\tilde{p}$, of sampling an augmentation in each group (prediction head), so the real absolute threshold value will be, e.g., $l \cdot \tilde{p}$. Taking an example of the Crop prediction head with 16 (1+15) magnitudes in total, $\tilde{p} = 1/16$. Hyperparameters are optimized using grid search.

| | CIFAR10 | | | CIFAR100 | Imagenette | ImageNet | SVHN |
|---|---|---|---|---|---|---|---|
| Model | WRN34-10 | WRN34-10 | ViT-B/4 | PRN18 | ViT-B/16 | ConvNeXt-T | PRN18 |
| Training | AT(-SWA) | AT-AWP(-SWA) | AT | AT | AT | AT | AT |
| Policy backbone | PRN18 | PRN18 | PRN18 | PRN18 | ViT-B/16 | PRN18 | PRN18 |
| Affinity model | WRN34-10 | WRN34-10 | ViT-B/4 | PRN18 | ViT-B/16 | ConvNeXt-T | PRN18 |
| $\lambda$ | 0.4, 0.2, 0.1 | 0.7 0.5 0.4 | 0.4 | 0.2 | 0.3 | 0.7 | 0.01 |
| $\beta$ | 0.8 | 0.8 | 0.8 | 0.8 | 0.8 | 2 | 0.3 |
| Diversity limits ($l$,$u$) | (0.9, 4.0) | (0.9, 4.0) | (0.8, 4.0) | (0.9, 4.0) | (0.8, 4.0) | (0.8, 4.0) | (0.7, 4.0) |
| Policy learning rate | 0.001 | 0.001 | 0.001 | 0.001 | 0.1 | 0.001 | 0.001 |

### F.4 CONFIGURATION OF COMPARED DA METHODS

AutoAugment was parameterized as in (Cubuk et al., 2019) since we did not have sufficient resource to optimize. For AutoAugment, augmentations were applied in the order of HorizontalFlip-RandomCrop-AutoAugment-Cutout (16x16) on CIFAR10 and Imagenette, and AutoAugment-Cutout (20x20) on SVHN, as in (Cubuk et al., 2019). TrivialAugment is parameter-free so no tuning was needed. For TrivialAugment, augmentations were applied in the order of HorizontalFlip-RandomCrop-TrivialAugment-Cutout (16x16) on CIFAR10 and Imagenette, and TrivialAugment-Cutout (16x16) on SVHN, as in (Müller & Hutter, 2021). For Cutmix, $\alpha = 0.25$ and $\beta = 1$ on CIFAR10 as optimized in (Li & Spratling, 2023c); $\alpha = 1$ and $\beta = 1$ on Imagenette and SVHN as suggested in (Yun et al., 2019). For Cutout, the size of cut-out area was 20x20 on all three datasets as in (Li & Spratling, 2023c). Cutout and Cutmix were applied with the default (baseline) augmentations in the order of HorizontalFlip-RandomCrop-Cutout and -Cutmix respectively on CIFAR10 and Imagenette, while no additional augmentations were applied on SVHN. For IDBH, IDBH[strong]-CIFAR10 was used on CIFAR10 and Imagenette, and IDBH-SVHN was used on SVHN.

We only compare our method against the baseline and AutoAugment on ImageNet. AutoAugment is selected because it is one of the two methods closest to AROID and has a pre-optimized version for ImageNet while the other closest work IDBH doesn't. Due to the tremendous cost of conducting AT on ImageNet and the limit of our computational resource, we can't optimize other DA methods for AT on ImageNet so they are not included to avoid unfair comparison. In fact, like most other researchers, we don't have enough time and resource to train all competitive DA methods even without re-optimization of hyperparameters.

### F.5 CONFIGURATION OF COMPARED STATE-OF-THE-ART ROBUST TRAINING METHODS

We only re-implemented the algorithms of SWA and AWP to report the result based on our runs, while the result of the others including MART, MART-AWP, SEAT, LAT-AT and LAS-AWP were copied directly from their original works except that the result of MART was copied from (Wu et al., 2020) for a better aligned training setting. SWA was implemented as in (Rebuffi et al., 2021) with a decay rate of $\tau = 0.999$. AWP was configured as in (Wu et al., 2020) with $\beta = 0.005$. Note that the same configurations of SWA and AWP were used to train with baseline DA and AROID.

## G ADDITIONAL RESULTS

### G.1 ROBUSTNESS EVALUATION WITH MORE ATTACKS

To further ensure our robustness evaluation is reliable, we additionally evaluate AROID and other related works using three more adversarial attacks PGD (Madry et al., 2018), CW (Carlini & Wagner, 2017) and JITTER (Schwinn et al., 2023) in Tab. 8. AROID is consistently superior under various adversarial attacks (AuoAttack, PGD, CW, JITTER).

Table 8: Robustness evaluation against more adversarial attacks. PGD uses 50 steps and 10 restarts. CW and JITTER use 100 steps. Note that the abnormally superior PGD robustness but worse against other attacks of Cutmix suggest a false security caused by obfuscated gradients.

| Augmentation | CIFAR10+WRN34-10 | | | | | Imagenette+ViT-B/16 | | | | |
|---|---|---|---|---|---|---|---|---|---|---|
| | Clean | AA | PGD | CW | JITTER | Clean | AA | PGD | CW | JITTER |
| baseline | 85.83 | 52.26 | 55.50 | 54.27 | 53.59 | 92.73 | 66.47 | 68.10 | 68.47 | 68.80 |
| Cutout | 86.95 | 52.89 | 55.35 | 55.02 | 54.61 | 93.27 | 67.20 | 68.40 | 68.67 | 69.40 |
| Cutmix | 86.88 | 53.38 | **60.13** | 56.98 | 56.47 | 93.87 | 70.20 | **73.10** | 71.80 | 72.20 |
| AutoAugment | 87.71 | 54.60 | 58.87 | 56.31 | 55.67 | 95.13 | 67.60 | 68.93 | 69.87 | 70.67 |
| TrivialAugment | 87.35 | 53.86 | 57.46 | 55.28 | 55.49 | **95.25** | 69.00 | 70.95 | 70.65 | 71.50 |
| IDBH | 88.61 | 55.29 | 58.27 | 57.36 | 56.96 | 95.20 | 69.93 | 70.20 | 70.80 | 71.67 |
| AROID (ours) | **88.99** | **55.91** | 59.68 | **58.18** | **57.63** | 94.88 | **71.32** | 71.80 | **72.80** | **73.12** |

Table 9: Comparison of various DA methods when trained by alternative AT methods like TRADES and SCORE for on CIFAR10 with PRN18. $\lambda$ is 0.6 for TRADES and 0.3 for SCORE. The other hyperparameters are configured by default as specified in Appendix F.

| DA Method | TRADES | | SCORE | |
|---|---|---|---|---|
| | Clean | AA | Clean | AA |
| RandomCrop | 83.01 | 49.10 | 80.19 | 48.88 |
| Cutout | 81.74 | 48.98 | 82.02 | 50.08 |
| AutoAugment | 80.76 | 48.64 | 81.68 | 49.93 |
| TrivialAugment | 80.91 | 48.04 | 80.39 | 49.49 |
| IDBH | 82.49 | 50.86 | 82.35 | 50.97 |
| AROID (ours) | **84.04** | **51.33** | **82.69** | **51.18** |

## G.2 GENERALIZATION TO OTHER AT METHODS

To further test the generalizability of AROID to alternative AT methods, we integrate AROID into two more superior AT methods TRADES (Zhang et al., 2019) and SCORE (Pang et al., 2022) in Tab. 9. AROID achieves highest accuracy and robustness among all the tested DA methods under both advanced AT methods. Overall, combining with the result in Section 4.2, AROID generalizes well to various AT methods (PGD, TRADES, SCORE, AWP, SWA).

AROID is combined with other AT methods in the same way as any other data augmentation: simply use the sampled data augmentation policy to augment the data before generating adversarial examples. The update of the policy model is independent of the adversarial training method used.

Table 10: Comparison of our method with more state-of-the-art AT methods on PRN18. The results of the compared methods are copied from their papers. "-" means that the corresponding result is not reported in the original work.

| Training | CIFAR10 | | CIFAR100 | | SVHN | |
|---|---|---|---|---|---|---|
| SCORE (Pang et al., 2022) | 83.75 | 49.57 | - | - | - | - |
| Consistency (Tack et al., 2022) | 84.65 | 47.83 | **60.21** | 23.71 | - | - |
| CFA (Wei et al., 2023) | 80.40 | 50.10 | - | - | - | - |
| HAT (Rade & Moosavi-Dezfooli, 2022) | 84.86 | 48.85 | 58.73 | 23.34 | 92.06 | 52.06 |
| UIAT (Dong et al., 2023) | **85.01** | 49.11 | 59.55 | 25.73 | 93.28 | 52.45 |
| FSR (Kim et al., 2023) | 84.49 | 48.45 | - | - | - | - |
| AROID (ours) | 84.68 | **50.57** | 60.17 | **26.56** | **93.30** | **54.49** |

Table 11: Comparison of the performance of various DA methods on SVHN with PRN18. The **best** and second best results are highlighted in each column. The baseline was no augmentation.

| DA Method | Accuracy | Robustness |
|---|---|---|
| baseline | 90.54±.74 | 47.56±.71 |
| Cutout | 90.69±.51 | 50.88±.45 |
| Cutmix | 91.13±.25 | 51.95±.40 |
| AutoAugment | 93.68±.17 | 54.15±.06 |
| TrivialAugment | 93.44±.37 | 52.78±.26 |
| IDBH | **93.70±.13** | **54.56±.29** |
| AROID (ours) | 93.65±.13 | 54.35±.10 |

Table 12: The performance of various DA methods on CIFAR10 with WRN34-10 when incorporating Stochastic Weight Averaging (SWA).

| DA Method | Accuracy | Robustness |
|---|---|---|
| baseline | 84.30 | 54.29 |
| Cutout | 87.26 | 55.38 |
| Cutmix | 86.78 | 55.60 |
| AutoAugment | 85.48 | 54.89 |
| TrivialAugment | **88.03** | 56.03 |
| IDBH | 87.48 | 56.45 |
| AROID (ours) | 87.84 | **56.67** |

### G.3 COMPARISON TO MORE STATE-OF-THE-ART AT METHODS

To further demonstrate the superiority of AROID over the existing works, we compare AROID with more state-of-the-art AT methods in Tab. 10. AROID outperforms all of them regarding robustness across various datasets further confirming the superiority of our method.

### G.4 COMPARISON TO OTHER DA METHODS ON MORE DATASETS

On SVHN our method achieves a similar accuracy and robustness to the best existing method, IDBH as shown in Tab. 11. The slightly better performance of IDBH on SVHN compared to AROID is likely due to the hyperparameters for IDBH having been tuned on the target model, whereas the hyperparameters for IDBH on the other datasets were tuned on a simplified proxy model (PRN18) due to the computational cost of using the target model. This highlights an advantage of AROID over IDBH: an improved efficiency leading to a more effective DA policy tuned to the target network (discussed in Section 4.5).

### G.5 ABLATION STUDY

This section verifies the sensitivity of our method to its hyperparameters and several design choices. The experiments were conducted on CIFAR10 with PRN18 and Imagenette with ViT-B/16 using the setup specified in Appendix F. The default values of hyperparameters are the ones marked color green in Fig. 3.

#### G.5.1 HYPERPARAMETERS

**Policy update frequency** $K$. Figs. 3i and 3l show that the highest accuracy and robustness were achieved when $K = 5$, i.e., the lowest frequency under the test. This implies that AT benefits from a more "up-to-date" DA. Furthermore, it seems possible to trade accuracy for efficiency by setting a larger $K$ (up to 20) while maintaining similarly high robustness. In general, the accuracy and robustness of our method declines with lower policy update frequency.

**Number of trajectories** $T$. Figs. 3h and 3k show that high accuracy and robustness are achieved around $T = 8$. This suggests that (1) there is a minimum requirement on the amount of trajectories

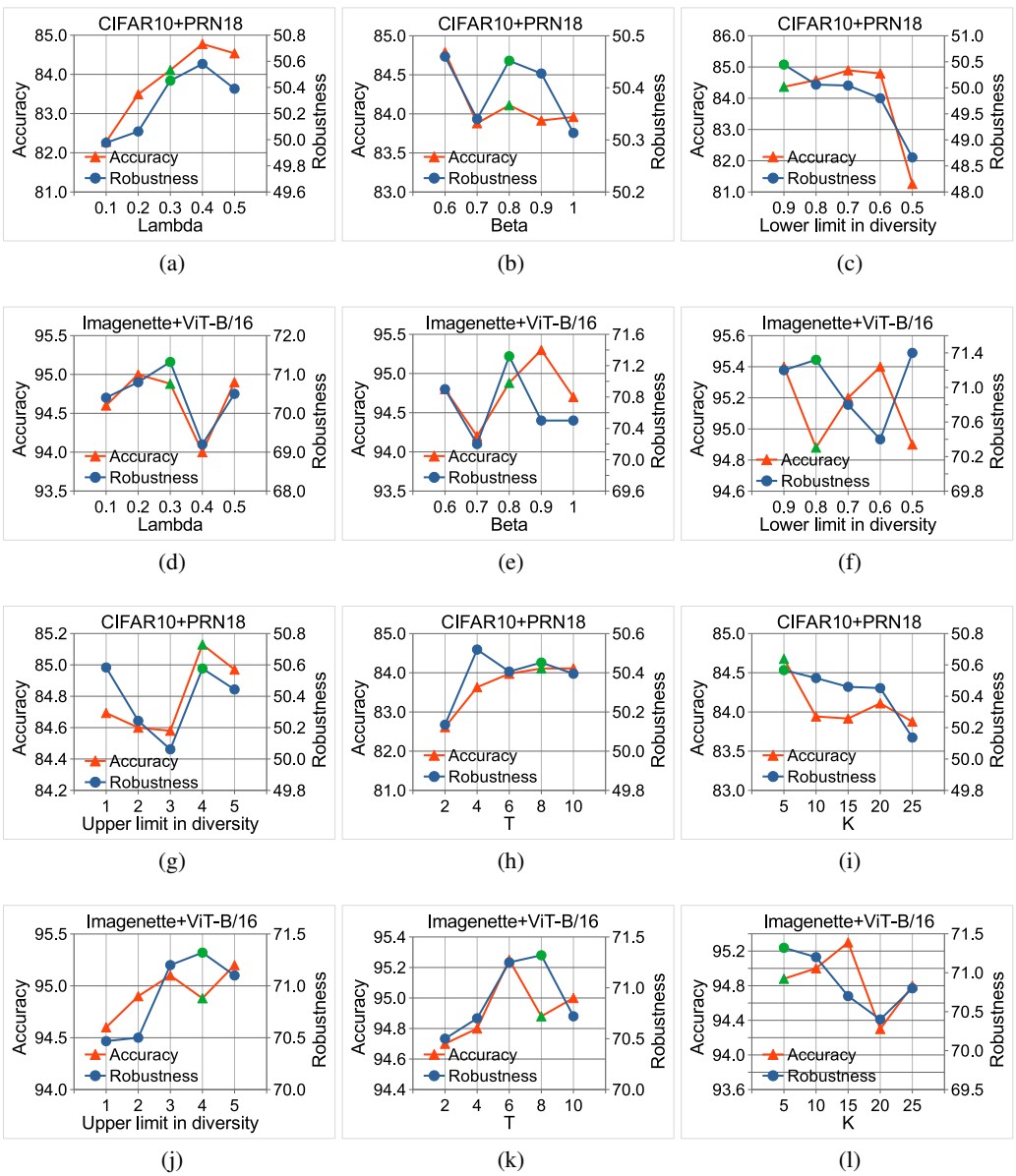

Figure 3: Ablation study of hyper-parameters $\lambda$, $\beta$, $l$, $u$, $T$ and $K$ for CIFAR10 with PRN18 (even rows) and Imagenette with ViT-B/16 (odd rows). The selected value for each hyper-parameter is marked green color.

Table 13: Comparison of the various policy model backbone architectures on CIFAR10 with a target model of PRN18.

| Model | Size (M) | Clean | AA |
|-------|----------|-------|-----|
| WRN10-1 | 0.08 | 84.16 | 50.25 |
| WRN22-1 | 0.27 | 84.32 | **50.57** |
| WRN34-1 | 0.47 | **84.73** | 50.38 |
| WRN70-1 | 1.05 | 84.04 | 50.28 |
| PRN18 | **11.17** | 84.68 | **50.57** |

Table 14: Comparison of uniform sampling from AROID DA space on CIFAR10 with PRN18.

| DA | Clean | AA |
|----|-------|-----|
| baseline | 82.50 | 48.21 |
| Uniform | 81.00 | 49.18 |
| AROID | **84.68** | **50.57** |

for our policy gradient estimator to be accurate and (2) our method may not benefit from increasing $T$ beyond 8.

**Strength of Vulnerability** $\lambda$. As shown in Figs. 3a and 3d, robustness first increases and then decreases within the tested range of value. This is consistent with the prior that AT benefits from appropriate hardness but degrade if data augmentations are overly hard (Li & Spratling, 2023c).

**Strength of Diversity** $\beta$. As shown in Figs. 3b and 3e, the performance within the tested range of value is close. We then further test the effect of removing Diversity by setting $\beta = 0$ on CIFAR10 with PRN18. We observed that accuracy drops to 73.88% from 84.68% and robustness drops to 22.47% from 50.57%. Training the policy network failed without Diversity as the output policy distribution concentrated on several sub-policies, i.e., gave zero probabilities at the remaining sub-policies. The REINFORCE method failed to recover from this situation because it no longer explored other opportunities. Overall, these suggest that having a certain strength of Diversity constraint is important for our policy learning, but no clear benefit is observed as the constraint is further enhanced by increasing beta within the tested value range.

**Summary**. We observe that, within the tested value range, hyper-parameters like $\lambda$, $\beta$, $T$ and $K$ have a quite similar trend in both settings, while the lower limit $l$ (Figs. 3c and 3f) and upper limit $u$ (Figs. 3g and 3j) in the diversity objective shows slightly different trends between the two settings. Despite the slightly different behaviors of a few hyper-parameters, the optimal value of hyper-parameters is observed to transfer across these two settings, i.e., they achieve reasonably good performance with a similar set of hyper-parameter values $T = 8$, $K = 5$, $l = 0.8/0.9$, $u = 4$, $\lambda = 0.3$, $\beta = 0.8$. We also find this setting transfers well across different AT methods of PGD, SCORE and TRADES since we can only tune the value of $\lambda$ while keep the rest unchanged to achieve reasonably good performance and outperform the other compared data augmentations.

### G.5.2 POLICY MODEL ARCHITECTURE

Interestingly, we observed in Tab. 13 that for CIFAR10 a relatively small model WideResNet10-1 (a WideResNets with depth 10 and widening factor 1) with 0.08M parameters is sufficient for learning the DA policy for a relatively large target model PRN18 with 11.17M parameters and further increasing capacity beyond this scale, even 100x, does not benefit either accuracy or robustness. Therefore, the policy model can be much smaller than the target model.

### G.5.3 UNIFORM SAMPLING

We experimented where of uniformly sampling data augmentations are uniformly sampled from AROID's data augmentation space for AT, dubbed Uniform in Tab. 14. As shown in the table, AROID significantly improves accuracy and robustness over its uniformly sampled counterpart suggesting the necessity of optimizing data augmentation policy.

### G.6 COMPARISON AGAINST SOTA IMAGENET RESULTS

We notice that a concurrent work (Singh et al., 2023) has achieved a higher record of robustness for ConvNeXt-T on ImageNet. We believe that directly comparing our result against (Singh et al., 2023) is unfair. The experiment setting of (Singh et al., 2023) is substantially different from ours and several techniques they used are advantageous. For example, they use a stronger attack APGD compared

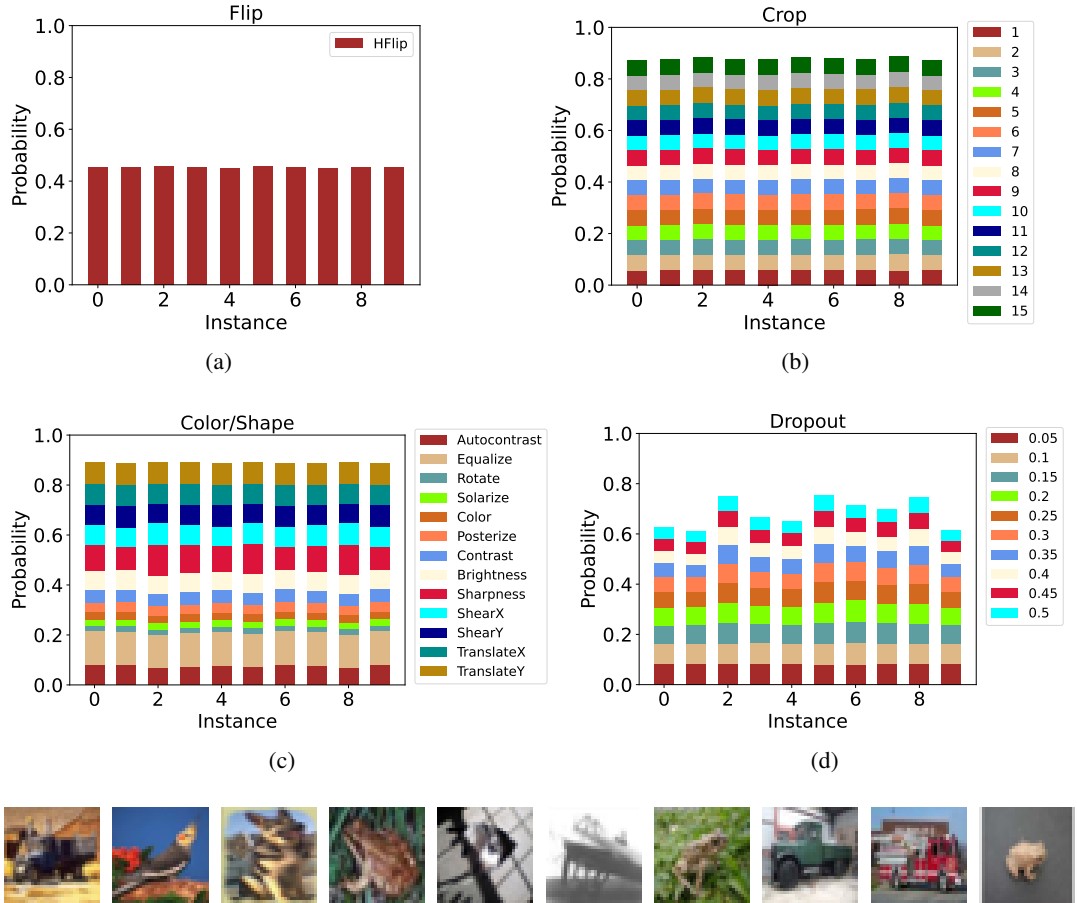

Figure 4: Visualization of the learned DA policies, applied to ten images randomly sampled from CIFAR10 training set, for the Flip, Crop, Color/Shape and Dropout types of augmentations. The policy model is resumed from a checkpoint saved at the end of $110^{th}$ epoch, which is randomly sampled from 200 epochs, i.e., a course of training a WRN34-10 model on CIFAR10 (following the training setting as specified in Appendix F). The sampled ten images are visualized at the bottom in an order of x-axis in the above figures. The chance of applying no transformation (Identity) is the gap between the colored bar and the top (i.e., score of 1.0). In the Color/Shape group, the probabilities of different magnitudes are not shown separately, but are summed to get the overall probability of a transformation.

to our naïve PGD to generate training adversarial examples. They use EMA and label smoothing to enhance the performance. The batch size is 1392 while ours is 128, which has been observed to benefit generalization on ImageNet. The optimizer and learning rate schedule are AdamW and cosine decay which are different from our SGD and multi-step decay.

Instead of comparing AROID against the heavy data augmentation of (Singh et al., 2023), we find it may be more useful to apply AROID to optimize it. The heavy data augmentation is composed of several individual data augmentation methods, RandAugment, CutMix, MixUp and Random Erasing, and each of them has a few hyperparameters jointly constituting a large search space. As we learn from (Singh et al., 2023), they did not sufficiently search this space to optimize the performance. We consider this as an opportunity to apply our method to automatically and efficiently optimize these hyperparameters to improve on the original scheme.

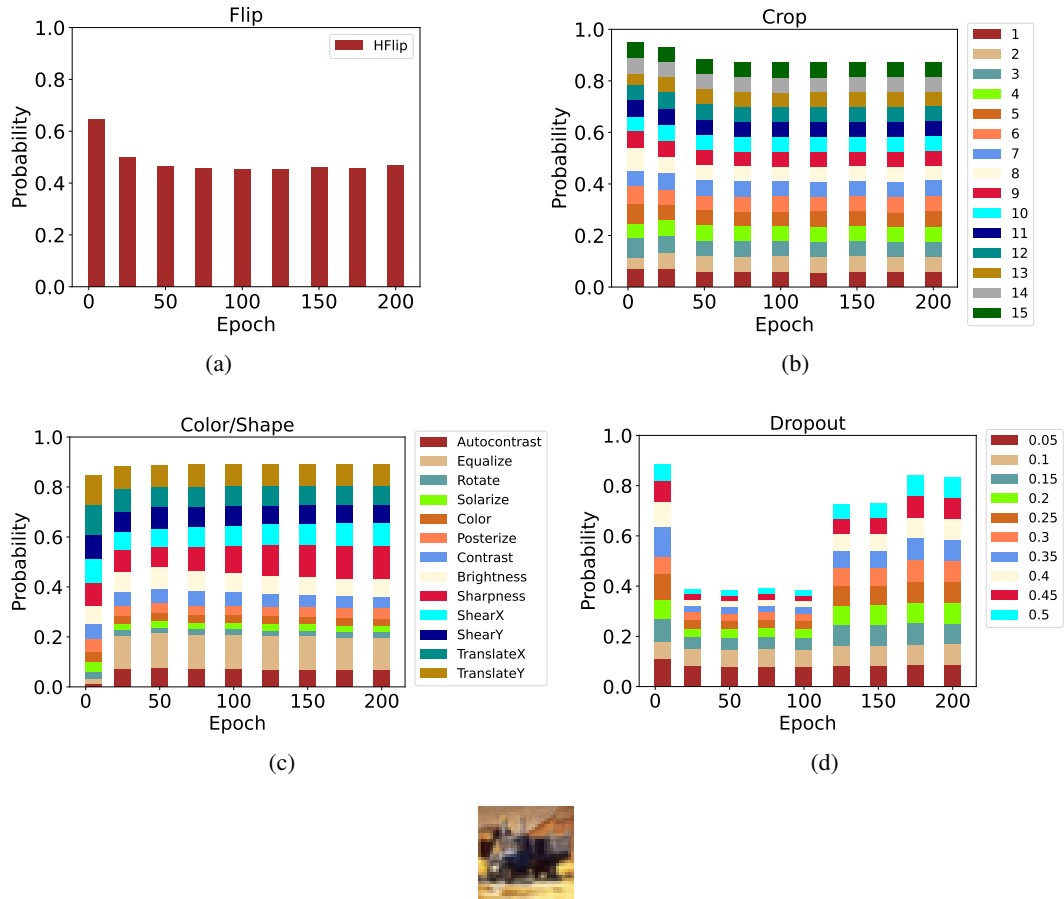

Figure 5: Visualization of how the learned DA policies evolve as training progresses. The same, randomly sampled, image (visualized at the bottom) was used across epochs (5, 25, 50, 75, 100, 125, 150, 175, 200) to produce the policies. The first bar in each sub-figure corresponding to the epoch 5 describes the initial status of the policy model (recalling that the training of policy model starts from epoch 5). For each bar in the figures, the policy model was resumed from the checkpoint saved at the corresponding epoch (x-axis) in the same course of training. The chance of applying no transformation (Identity) is the gap between the colored bar and the top (i.e., the score of 1.0). In the Color/Shape group, the probabilities of different magnitudes are not shown separately, but are summed to get the overall probability of a transformation.

## H    VISUALIZATION OF LEARNED DA POLICIES

Fig. 4 visualizes the learned distribution of DAs for different, randomly sampled, data instances. Instance-wise variation of the learned DA policy is visible for the Color/Shape augmentations (Fig. 4c) and evident for the Dropout augmentations (Fig. 4d), but subtle in the rest (Fig. 4a and Fig. 4b). Note that even for the different data instances from the same class (e.g., instances 4, 7, 10 from the class "frog"), the learned DA distributions can still differ considerably (Fig. 4d). This confirms that (1) AROID is able to capture and meet the varied demand of augmentations from different data instances, and (2) such demand exists for some, but not all, augmentations. These observations may explain why many instance-agnostic DA methods such as IDBH, despite being inferior to ours, still work reasonably well (see Tab. 1).

It was also observed in Fig. 5 that the learned DA policy for the same data instance evolved as training progressed. In the Color/Shape group (Fig. 5c), augmentations like Sharpness became observably more likely to be selected while others such as ShearY became less probable as training continued. Dropout (i.e. Erasing; Fig. 5d) particularly with large magnitudes was rarely applied prior to 100th epoch, i.e., the first decay of learning rate. The possibility of applying Crop (i.e. Cropshift; Fig. 5b) and Flip (i.e. HorizontalFlip; Fig. 5a) first dropped until the first decay of learning rate and then stayed nearly constant afterwards.

Consistent to the previous findings on ST (Cubuk et al., 2019) and harmful augmentations (Rebuffi et al., 2021), we observed that AT on CIFAR10 favored mostly color-based augmentations like Equalize and Sharpness and disfavored geometric augmentations like Rotate and harmful augmentations like Solarize and Posterize (see both Fig. 4c and Fig. 5c). This verifies the effectiveness of our DA policy learning algorithm.

