# OpenReview forum: "AROID: Improving Adversarial Robustness through Online Instance-wise Data Augmentation"
_ICLR.cc/2024/Conference — ICLR 2024 Conference Withdrawn Submission_

### Official Review · Reviewer_aRvv · 2023-10-28

**Soundness:** 3 good
**Presentation:** 2 fair
**Contribution:** 3 good
**Rating:** 5
**Confidence:** 4

**Summary:**

The authors propose a novel adversarial training by introducing a policy network that selects, for each instance, the data augmentation procedure that could, potentially, maximize the robustness of the model. The experimental results show the algorithm is able to achieve good results compared with the current state-of-the art.

**Strengths:**

**originality**: The main contribution of this paper is the addition of a policy network that selects the best data augmentation procedure to maximize the network's robustness. The idea is interesting, and seems to be novel.

**quality**:  The experimental section shows promising results against current state-of-the-art techniques.

**clarity**: The idea is sound, and the authors put high emphasis in describing the motivation of each decision made, specially regarding to the loss terms.

**significance**: works that target network's robustness with lossless accuracy are of special interest for the community.

**Weaknesses:**

**originality**: Besides the policy network, the rest of the paper is a combination of well-known techniques. Although, I must say they are combined in a sound and clever way.

**quality**: The experimental section results are confusing. If I understood it correctly, the baseline results consider the network training without any adversarial techniques. Thus, it is known that the accuracy of either WRN and ViT algorithms are higher than those provided by the authors (over 95% in CIFAR10 and 80% in CIFAR100 for the WRN network, for instance). I also think Table 2 should include the training  time of every state-of-the-art algorithm.

**clarity**: Some hyper-parameter default values appear to be missing ($l$ and $u$ bounds in the Diversity penalty. I suggest the authors to include all hyper-parameter values in the main paper.

**significance**: Due to the experimental section problem, it is not possible to conclude that this algorithm can provide a robustness increase with lossless accuracy performance.

**Questions:**

- Why the baseline methods have lower accuracy scores than the ones provided by the authors of the network models?
- The diversity penalty (Eq. 7) seems to be quite overcomplicated. If the authors aim to obtain values close to an uniform distribution, should the authors use a more usual entropy loss? $$\mathcal{L}^h_{div}(\mathbf{x}) = \sum_i p^h_i(\mathbf{x}; \mathbf{\theta}) \log(p^h_i(\mathbf{x}; \mathbf{\theta})) $$
- Can the authors provide an ablation study regarding to the different penalties applied to the policy network?

---

> ### Author Response · Authors · 2023-11-21
>
> > originality: Besides the policy network, the rest of the paper is a combination of well-known techniques. Although, I must say they are combined in a sound and clever way.
> >
>
> The novelty of this work lies in the idea of online instance-wise data augmentation and the proposal of three policy objectives to allow efficient and effective learning of data augmentation policies for adversarial robustness.
>
> > quality: The experimental section results are confusing. If I understood it correctly, the baseline results consider the network training without any adversarial techniques. Thus, it is known that the accuracy of either WRN and ViT algorithms are higher than those provided by the authors (over 95% in CIFAR10 and 80% in CIFAR100 for the WRN network, for instance).
> >
>
> Sorry for the missing details. The results in Table 1 are the performance of different data augmentation methods applied to adversarial training. The “baseline” in the table refers to the baseline data augmentation with adversarial training instead of their standardly-trained counterparts.
>
> > I also think Table 2 should include the training time of every state-of-the-art algorithm.
> >
>
> Thanks for your suggestion. However, we can only provide now the training time of SWA and AWP as we have implemented them by ourselves, while the results of the remaining SOTA methods are copied from their papers and we do not have a working implementation in our local device to report the training time. The cost of training a PreActResNet18 on CIFAR10 with SWA and AWP for 200 epochs is 8.85 and 6.72 hours respectively on A100 GPU. If the reviewer has any method of interest, we are happy to try to implement it to report.
>
> > significance: Due to the experimental section problem, it is not possible to conclude that this algorithm can provide a robustness increase with lossless accuracy performance.
> >
>
> We wish the above clarification could clear your confusions so that you could re-assess our work. We also would like to point you to our new result of training with extra real data in [shared response](https://openreview.net/forum?id=ufZp6pvOvE&noteId=J7xQVCLpSt).
>
> > clarity: Some hyper-parameter default values appear to be missing (l, u and bounds in the Diversity penalty. I suggest the authors to include all hyper-parameter values in the main paper.
> >
>
> Thanks for your suggestion. The values of all hyperparameters are specified in Table 7 with a description in appendix F.2. We have added a short description and reference to the appendix at the beginning of section 4.
>
>
>
> > The diversity penalty (Eq. 7) seems to be quite overcomplicated. If the authors aim to obtain values close to an uniform distribution, should the authors use a more usual entropy loss?
> >
>
> Thanks for your suggestion. The point of proposing Eq. 7 is to enforce a relaxed uniform distribution to only penalize overly small ($< l$) and large ($> u$) probabilities. This enables a more flexible distribution between $l$ and $u$. Besides, Eq. 7 is quite straightforward instead. Mathematically, it just averages the signed logprobability of logits outside the range.
>
> > Can the authors provide an ablation study regarding to the different penalties applied to the policy network?
> >
>
> The strength of penalties on hardness and diversity is mainly controlled by the hyper-parameters lambda and beta respectively. In summary, our method benefits from an appropriate level of hardness and requires a necessary level of diversity to prevent from collapse. Please refer to appendix G.5.1 with result figures and detailed analysis. Appendix G.5 also conducts the ablation study on other hyperparameters and the design choices of our method. We are happy to provide if anything else you are interested in.

---

> > ### Comment · Reviewer_aRvv · 2023-11-22
> >
> > Dear authors,
> >
> > I would like to thank the authors for the extensive response. However, I still have concerns regarding to the results provided in the Shared response. The authors claimed that comparing their algorithm against BDM is not fair because of the extra data. As the BDM code is publicly available, I suggest the authors to use it and to provide a fair experiment that can compare both techniques. I would keep my initial score. I think the idea is interesting, but more work has to be done before publishing it.

---

> ### Author Response · Authors · 2023-11-22
>
> Dear Reviewer,
>
> thanks for your reply. We would like to emphasize that **AROID is a data augmentation method instead of an "extra data" method so it does not have to beat BDM**. The key point is that **AROID is complementary to training with extra data, i.e., by incorporating extra data into AROID the accuracy and robustness are further improved**. This has been confirmed by the results in shared response. We have also conducted extensive experiments across varied datasets, models and adversarial training methods in the paper to show the superiority of AROID over all competitive data augmentation methods regarding accuracy, robustness and policy search efficiency. We hope you could reconsider your assessment of our work based on the above information.
>
> We are absolutely happy to and will add the result of combining AROID with BDM in the next revised version but it is just too time-consumsing to finish by rebuttal deadline.

---

### Official Review · Reviewer_tKpP · 2023-10-30

**Soundness:** 2 fair
**Presentation:** 3 good
**Contribution:** 2 fair
**Rating:** 5
**Confidence:** 5

**Summary:**

This paper proposes an automated data augmentation method, AROID, to address the generalization issue in adversarial training. Specifically, the authors argue that tailored image transformations should be applied for each training sample and at each stage of training. They introduce the periodic learning of the data augmentation policy model during adversarial training. Based on the results of this policy model for each image, combinations of image transformations are applied during adversarial training. The experiments demonstrate the efficiency and effectiveness of the proposed method compared to other automated data augmentation techniques.

**Strengths:**

1. This paper is overall clearly clarified and well organized.
2. The proposed method is experimentally demonstrated to be more efficient and effective when compared to other automated data augmentation techniques.

**Weaknesses:**

1. The proposed method involves many hyperparameters (u, l, beta, lambda, K, policy model hyperparams), and the discovered policy leads to compromised results in different training settings, thereby significantly increasing the cost of adversarial training.
2. It is expected that learning an instance-wise data augmentation policy model would be challenging as the data distribution expands, yet there is no analysis provided on this aspect.
3. Results for CIFAR100+WRN34-10 or ViT-B/16 are missing from Table 1.
4. Rebuffi et al. proposed the use of Cutmix in conjunction with SWA. Therefore, it is necessary to present the results incorporating SWA in Table 1.
5. It is necessary to present the results when used in conjunction with TRADES (Zhang et al., 2019).
6. AutoAugment is a method proposed without considering adversarial training. Therefore, using pre-trained AutoAugment as is would be unfair.
7. It is required to show the results when such approaches are applied alongside the use of more training data [1,2].

[1] Carmon, Yair, et al. "Unlabeled data improves adversarial robustness." NeurIPS 2019.
[2] Gowal, Sven, et al. "Improving robustness using generated data." NeurIPS 2021.

**Questions:**

(Copy from Weaknesses)
1. Results for CIFAR100+WRN34-10 or ViT-B/16 are missing from Table 1.
2. Rebuffi et al. proposed the use of Cutmix in conjunction with SWA. Therefore, it is necessary to present the results incorporating SWA in Table 1.
3. It is necessary to present the results when used in conjunction with TRADES (Zhang et al., 2019).
4. It is required to show the results when the proposed method is applied alongside the use of more training data.

---

> ### Author Response · Authors · 2023-11-21
>
> > The proposed method involves many hyperparameters (u, l, beta, lambda, K, policy model hyperparams), and the discovered policy leads to compromised results in different training settings, thereby significantly increasing the cost of adversarial training.
> >
>
> The cost of training does not significantly increase due to hyperparameters because most of our hyperparameters transfer well across different settings so that only a few need to be tuned for good performance. In practice, across various datasets, model architectures and adversarial training methods experimented in this work, a decent performance surpassing the competitive methods can be achieved by searching lambda within [0, 1] in a step size of 0.1 in most cases. This has been discussed in appendix E.2.
>
> > It is expected that learning an instance-wise data augmentation policy model would be challenging as the data distribution expands, yet there is no analysis provided on this aspect.
> >
>
> We have experimented with two pairs of datasets, CIFAR10-CIFAR100 and Imagenette-ImageNet (Tab.1 and Tab.3 in our paper), which covers the expansion of data distribution in the number of classes, input resolution and the number of training data. We now add a new experiment of training with extra real data [1] to further capture the expansion of data distribution. Our method consistently achieves the best performance over the competitive methods in all above settings, confirming the effectiveness of our method under the expansion of data distribution.
>
> > Results for CIFAR100+WRN34-10 or ViT-B/16 are missing from Table 1.
> >
>
> The results have been added to our paper in Table 1. As shown below, our method consistently outperforms others regarding both accuracy and robustness.
>
> | DA Method | Accuracy | Robustness |
> | --- | --- | --- |
> | baseline | 61.44 | 27.98 |
> | Cutout | 59.04 | 27.51 |
> | Cutmix | 58.57 | 27.49 |
> | AutoAugment | 64.10 | 29.08 |
> | TrivialAugment | 62.55 | 28.97 |
> | IDBH | 60.93 | 29.03 |
> | AROID (ours) | 64.44 | 29.75 |
>
> > Rebuffi et al. proposed the use of Cutmix in conjunction with SWA. Therefore, it is necessary to present the results incorporating SWA in Table 1.
> >
>
> Thanks for your suggestion. Due to the time, we have only got the results of incorporating SWA for CIFAR10 with WRN34-10. The results have been added to our paper in Table 13. As shown below, our method still achieves the highest robustness among all competitive DA methods.
>
> | DA Method | Accuracy | Robustness |
> | --- | --- | --- |
> | baseline | 84.30 | 54.29 |
> | Cutout | 87.26 | 55.38 |
> | Cutmix | 86.78 | 55.60 |
> | AutoAugment | 85.48 | 54.89 |
> | TrivialAugment | 88.03 | 56.03 |
> | IDBH | 87.48 | 56.45 |
> | AROID (ours) | 87.84 | 56.67 |
>
> > It is necessary to present the results when used in conjunction with TRADES (Zhang et al., 2019).
> >
>
> Yes, we did include the result of different data augmentation methods on TRADES as well as SCORE [3] in Tab. 9 in appendix G.2. Our method consistently outperforms other data augmentation methods regarding both accuracy and robustness on these alternative adversarial training methods.
>
> > AutoAugment is a method proposed without considering adversarial training. Therefore, using pre-trained AutoAugment as is would be unfair.
> >
>
> One of the serious limitation of AutoAugment is its prohibitive cost of policy search. As shown in Tab. 5 in our paper, learning policy towards clean accuracy based on standard training as naively done by AutoAugment requires 5000 hours on a P100 GPU. Theoretically, if standard training is replaced by adversarial training to learn policy for adversarial robustness, the cost would be 10 times more assuming an usual 10-step adversarial training used. This is so time-consuming that we cannot afford to re-optimize AutoAugment.
>
> > It is required to show the results when such approaches are applied alongside the use of more training data [1,2].
> >
>
> Thanks for your suggestion. Due to the time, we have only experimented [1] with our method. Below is the result on CIFAR10 with WRN-34-10 for 200 epochs and  a batch size of 128, while we present the result of a more advanced training set-up in [shared response](https://openreview.net/forum?id=ufZp6pvOvE&noteId=J7xQVCLpSt). As shown below, our method consistently achieves a large improvement over the baseline method RandomCrop. This discussion has been added to our paper.
>
> | Data Augmentation Method | Accuracy | Robustness |
> | --- | --- | --- |
> | RandomCrop | 88.78 | 57.95 |
> | AROID | 91.13 | 61.06 |
>
> Reference:
>
> [3] Pang et al., Robustness and Accuracy Could Be Reconcilable by (Proper) Definition, ICML 2022

---

> ### Comment · Reviewer_tKpP · 2023-11-23
>
> Thank you for addressing my concerns. I have one more question:
> 1. What is the reason that the newly added baseline results do not match the results shown in their original experiments? For example, Rebuffi et al. claimed that the CutMix+WA model achieved an Accuracy/Robustness of 86.18%/58.09% [1], and according to [2], Carmon et al. achieved an Accuracy/Robustness of 89.69%/59.53% using WRN-28-10.
>
>
> [1] Fixing Data Augmentation to Improve Adversarial Robustness, NeurIPS 2021.
> [2] https://github.com/fra31/auto-attack

---

> > ### Author Response · Authors · 2023-11-23
> >
> > Dear Reviewer,
> >
> > thanks for your reply! it's very nice to hear from you. We believe that the performance difference results from the fact that our training set-ups are different from theirs. It has been observed before in [3] that the performance of adversrial training is much sensitive to its training set-up. Specifically, [1] used TRADES to train for 400 epochs with a batch size of 512 and the model is Swish/SiLU variant of WideResNets, while ours is PGD10 AT for 200 epochs, 128 batch size and ReLU variant. Each of these changes has been observed before to benefit robustness. An ablation study on epochs and batch size is shown in [shared response](https://openreview.net/forum?id=ufZp6pvOvE&noteId=J7xQVCLpSt) where the robustness of our method increases by 1.1% through increasing epochs to 400 and batch size to 512. If we compensate all these measures, the result should be close to the one of [1]. For Carmon et al., there is also a difference in the adversarial training methods, learning rate schedules, etc. Nevertheless, even ignoring this, AROID with extra data still outperforms Carmon et al. from [2] by a clear margin. We hope this could clear your concern. Please let us know if anything else we could help.
> >
> > [3] Pang et al., Bag of Tricks for Adversarial Training, ICLR 2021

---

### Official Review · Reviewer_7wWe · 2023-10-30

**Soundness:** 2 fair
**Presentation:** 3 good
**Contribution:** 2 fair
**Rating:** 3
**Confidence:** 4

**Summary:**

This paper presents AROID, an adaptive Data Augmentation (DA) method to improve adversarial robustness while avoiding overfitting during Adversarial Training (AT). A common limitation of AT is indeed that it suffers from overfitting and data augmentation is a typical way to overcome this. However, existing DA methods for adversarial robustness have two limitations: 1) they require computing an optimal DA policy, which is expensive as each trial requires an AT run; 2) the found policy is equally applied to all training examples. This paper suggests that there is an opportunity to optimize the policy at each data point, which could lead to lower computational cost and better effectiveness. Evaluation on CIFAR-10, CIFAR-100 and ImageNet reveals that AROID is competitive with other DA methods (and outperforms more of them) while being more efficient than previous similar methods (accuracy-wise) by 3 orders of magnitude.

**Strengths:**

The paper presents an interesting idea in optimizing DA policy at instance-level rather than globally for all samples.

AROID can be used in offline mode, in which the policy has been previously optimized on a subset of data or on a smaller model.

The paper proposes a thorough experimental protocol that also includes an ablation study, demonstrating the contribution of all components (including the three objectives) to the effectiveness of AROID.

The paper does not point to a replication package to reproduce the results.

**Weaknesses:**

The paper makes strong overclaims. For example, I disagree that "AROID is the first automated DA method specific to adversarial
robustness"; or that "AROID achieves state-of-the-art robustness for DA methods on the standard benchmarks".

Indeed, standard evaluation benchmarks like RobustBench (https://robustbench.github.io/#leaderboard) include many robustness methods that rely on data augmentation and many of them outperforms AROID in terms of raw robust accuracy. I appreciate the fact that this may come from some controlled variables, like the used model architectures and parameters, but the lack of comparison (even qualitative) with these established leaderboards/benchmarks makes it difficult to assess the benefits of AROID compared to the available body of knowledge (beyond the few DA methods considered in the paper). More generally, the choice of baselines to compare to lacks proper justification.

Even considering only the baselines reported in the paper, the improvements in robust accuracy are limited to up to 3 percentage points compared to other DA methods (often less than this). Given the number of factors at play when determining empirical robustness, it is hard to establish AROID offers generalizable improvements.

The paper is also unclear in its efficiency evaluation. Indeed, Table 5 reports the time needed by AROID (in online or offline mode?) and other policy optimization methods during the policy search process only. Elsewhere in the paper, it is reported that AROID adds up 43% more time compared to standard AT whereas the "state-of-the-art" AT method named LAS-AT adds up to 52%. First, I would like to understand what is the practical cost of using AROID versus, e.g., IDBH; is this cost limited to policy search? I would think that in online mode, AROID has an additional cost that adds up at each instance (whereas IDBH computes the policy once and for all). Second, 43% more than standard AT is not a minor increase given the already huge cost of AT.

Overall, I have difficulties agreeing that AROID makes a significant step towards effective and efficient adversarial robustness methods, especially given the lack of justification for the chosen comparison baselines and the fact that there exists methods that improve robustness more significantly.

**Questions:**

- Can you elaborate on how AROID compares with the leading robustness methods reported in RobustBench?

- Table 5 reports the efficiency of AROID compared to similar methods for searching the search space only. Meanwhile, other numbers report that AROID requires 43% more time compared to standard AT while the other methods require 52% more than AT. Therefore, I wonder what is the total computational cost of using AROID versus the other DA methods if one was to using those in practice?

- In Table 5, have you used AROID in offline or online mode?

---

> ### Author Response · Authors · 2023-11-21
>
> > The paper makes strong overclaims. For example, I disagree that "AROID is the first automated DA method specific to adversarial robustness". Indeed, standard evaluation benchmarks like RobustBench include many robustness methods that rely on data augmentation
> >
>
> To our best knowledge, there is no existing automated data augmentation methods specifically proposed for adversarial training to boost adversarial robustness. The community did use many data augmentation methods like AutoAugment, Cutmix etc., which we have thoroughly reviewed in section 2 and compared against in Table 1 in our paper. However, they are all originally proposed for standard training to improve clean accuracy instead of adversarial training to improve robustness. Our claim emphasizes the point “specific to adversarial training” as it is the lack of adapting augmentation methods to adversarial training that leads to suboptimal robustness [2].
>
> > or that "AROID achieves state-of-the-art robustness for DA methods on the standard benchmarks"
> >
>
> This conclusion is made based on the results in Table 1 where our method outperforms all competitive methods regarding both accuracy and robustness across varied datasets and model architectures. The compared DA methods are selected to include common baselines, popular and state-of-the-art DA methods, and cover both heuristic and automated DA methods.
>
> On the other hand, the reviewer’s concern seems to be that some competitive DA methods have achieved records of higher robustness in the corresponding leaderboard of RobustBench. However, directly comparing our results against the results from RobustBench is, as agreed by the reviewer, unfair because of the uncontrolled variables like model architecture, adversarial training methods and training set-up, etc. For example, it is unfair to say that RandomCrop is better than AROID if the robustness of RandomCrop with WideResNet-70-16 is higher than that of AROID with ResNet18 since model size benefits robustness a lot. By contrast, we benchmark these DA methods in a standard (accepted by many previous works [1, 2, 3]), unified, setting with all other variables controlled to be same. We therefore believe our comparison is unbiased and our conclusion is correct.
>
> > Indeed, standard evaluation benchmarks like RobustBench (https://robustbench.github.io/#leaderboard) include many robustness methods that rely on data augmentation and many of them outperforms AROID in terms of raw robust accuracy. I appreciate the fact that this may come from some controlled variables, like the used model architectures and parameters, but the lack of comparison (even qualitative) with these established leaderboards/benchmarks makes it difficult to assess the benefits of AROID compared to the available body of knowledge (beyond the few DA methods considered in the paper). More generally, the choice of baselines to compare to lacks proper justification. Can you elaborate on how AROID compares with the leading robustness methods reported in RobustBench?
> >
>
> Kindly please refer to [shared response](https://openreview.net/forum?id=ufZp6pvOvE&noteId=J7xQVCLpSt) for comparison.
>
> > Even considering only the baselines reported in the paper, the improvements in robust accuracy are limited to up to 3 percentage points compared to other DA methods (often less than this). Given the number of factors at play when determining empirical robustness, it is hard to establish AROID offers generalizable improvements.
> >
>
> First, we would like to highlight that AROID also improves clean accuracy exhibiting a better trade-off between accuracy and robustness and, when compared to other automated DA methods, efficiency of policy learning.
>
> Second, we have conducted extensive experiments to verify the generalization of our method. Table 1 and 3 evaluates AROID across datasets, CIFAR10, CIFAR100, Imagenette and ImageNet, and model architectures, WRN34-10, ViT-B, PreActResNet18 and ConvNeXt-T. Table 2 and 9 (in appendix) evaluates AROID when combined with various adversarial training methods including TRADES, SCORE, SWA, AWP. [Shared response](https://openreview.net/forum?id=ufZp6pvOvE&noteId=J7xQVCLpSt) evaluates AROID when incorporating extra real data for training. Across all theses diverse settings, AROID demonstrates consistent and evident performance improvement regarding both accuracy and robustness. This suggests that AROID offers generalizable performance improvements.

---

> ### Author Response · Authors · 2023-11-21
>
> > In Table 5, have you used AROID in offline or online mode?
> >
>
> Online mode. We have added this explanation to our paper in Table 5.
>
> > The paper is also unclear in its efficiency evaluation. Indeed, Table 5 reports the time needed by AROID (in online or offline mode?) and other policy optimization methods during the policy search process only. Elsewhere in the paper, it is reported that AROID adds up 43% more time compared to standard AT whereas the "state-of-the-art" AT method named LAS-AT adds up to 52%. First, I would like to understand what is the practical cost of using AROID versus, e.g., IDBH; is this cost limited to policy search? I would think that in online mode, AROID has an additional cost that adds up at each instance (whereas IDBH computes the policy once and for all).
> >
>
> Efficiency of our method is discussed in section 3.4 and 4.5. The cost of automated DA methods consists of two parts: the policy search and the training of the target models. Offline methods like IDBH and AutoAugment separate these two parts, while online methods like AROID integrate both into one single run. **The cost of training target models is close for both classes of methods**, while AROID costs slightly more i.e. one more forward pass of policy model (much smaller than the target model) for sampling policy in each iteration. **Regarding policy search, AROID shows a clear efficiency improvement over IDBH and AutoAugment** as shown in Table 5. The advantage of IDBH and AutoAugment is that policy search can be an one-time cost if they are applied to train multiple models using the same setting (no need to re-optimize policy), whereas AROID has to repeat policy search for training every new model even using the same setting. However, when applied to a new training setting like a new dataset or model, IDBH and AutoAugment have to perform policy search to adapt their policies to the new setting for optimal performance. In this case the total cost is the time taken for policy search plus the time taken to train the target models. **Taking an example of CIFAR10 with PreActResNet18 model, AROID costs about 9.5 hours (K=5, T=8), while IDBH costs about 412.8 hours**. The prohibitive cost of policy search makes IDBH much less practical than ours to use. Last, **AROID can be also used in offline mode like IDBH to make its policy search a one-time cost.** As shown in Table 4, the offline AROID achieves 88.76% for accuracy and 55.61% for robustness on CIFAR10 with WRN34-10 still higher than the existing best method IDBH (88.61% for accuracy and 55.29% for robustness).
>
> > Second, 43% more than standard AT is not a minor increase given the already huge cost of AT.
> >
>
> First, this extra cost is mainly for policy search. The commonly used AutoAugment and IDBH cost dramatically more than ours for policy search (Table 5). Our method takes a significant step to reduce the cost of policy search making it practical to be used with adversarial training. Second, 43% is the cost of our most expensive setting (K=5, T=8). It can be reduced to around 10% by setting K=20 and T=4 with a slight cost of performance (Table 5). Note that even in this cheap setting AROID can still match the robustness of IDBH. Third, AROID can also be used in an offline mode with a slight cost of performance as discussed above.
>
> Reference:
>
> [1] Rice et al., Overfitting in adversarially robust deep learning, ICML 2020
>
> [2] Wu et al., Adversarial Weight Perturbation Helps Robust Generalization, NeurIPS 2020
>
> [3] Yu et al., Understanding Robust Overfitting of Adversarial Training and Beyond, ICML 2022
>
> [4] Carmon, Yair, et al. "Unlabeled data improves adversarial robustness." NeurIPS 2019.

---

### Official Review · Reviewer_Dov6 · 2023-11-02

**Soundness:** 3 good
**Presentation:** 1 poor
**Contribution:** 3 good
**Rating:** 6
**Confidence:** 3

**Summary:**

The paper presents an online adaptive data augmentation method called AROID for training a robust classification model. It is an extended method of the IDBH method, which can be viewed as a static method. The main idea is to use a different augmentation method for every input and this strategy evolves over training. The experiments show that the effectiveness of the proposed method.

**Strengths:**

The online adaptive idea is interesting. The efficiency of the method is much higher than previous data augmentation method including IDBH and AutoAugment. Experiments are extensive.

**Weaknesses:**

1. The description of the method is not clear.

The overall pipeline Fig. 1 is complicated, but lacks interpretation. In fact, some notations are not defined at all such as f_{aft}, f_{tgt} and f_{plc}. I can guess the meaning of the last two, but cannot guess the first one.

It is claimed that we need to perform bilevel optimization, alternate between target and policy models. I'm confused how the adversarial examples are obtained because to optimize eqn (9) we need go generate adversarial examples (see the function of rho). It seems that we have three optimization problems instead of two. Then how do we optimize them?

Table 2 reports the results of the proposed AROID method combined with different AT methods. But it is not described how they are combined.

2. The results on ImageNet are not convincing enough.

It seems the proposed method works well on small datasets. But the results on the large dataset ImageNet are not convincing enough. In ref [a], it is reported that ConvNeXt-T with random init., basic augment. + strong clean pre-training + heavy augmentations achieved 46.5% robust accuracy (see Table 1 in [a]). But this work only achieved 40.40% robust accuracy. It seems that the proposed data augmentation method is not as effective as the method in [a]. Anyway, I think it is necessary to compare with the method in [a].

[a] Naman D Singh, Francesco Croce, Matthias Hein, Revisiting Adversarial Training for ImageNet: Architectures, Training and Generalization across Threat Models. NeurIPS 2023 (arXiv:2303.01870).

**Questions:**

About the ImageNet experiment, if the experimental settings in [a] are used, e.g., run experiments upon (random init., basic augment. + strong clean pre-training), can AROID show advantage over heavy augmentations proposed in [a]?

---

> ### Author Response · Authors · 2023-11-21
>
> > The overall pipeline Fig. 1 is complicated, but lacks interpretation. In fact, some notations are not defined at all such as f_{aft}, f_{tgt} and f_{plc}. I can guess the meaning of the last two, but cannot guess the first one.
> >
>
> Thanks for your suggestion. The pipeline is described in detail in section 3. Regarding notations, $f_{aft}$, $f_{tgt}$ and $f_{plc}$ correspond to the affinity, target and policy model. Affinity model is a model pre-trained on the original data (i.e. without any data augmentation). It is used in the computation of Affinity metric for measuring distribution shift. For more details, please see section 3.2. We have added this explanation to Figure 1.
>
> > It is claimed that we need to perform bilevel optimization, alternate between target and policy models. I'm confused how the adversarial examples are obtained because to optimize eqn (9) we need go generate adversarial examples (see the function of rho). It seems that we have three optimization problems instead of two. Then how do we optimize them?
> >
>
> Bilevel optimization is for updating target and policy models. Strictly speaking, there are three optimization loops when concerning the generation of adversarial examples. The pseudo-code of the entire training pipeline is illustrated in Algorithm 1. The generation of adversarial examples and the update of the target model are performed as how they are in naive adversarial training. The only difference is that the training data now is augmented by the data augmentation sampled from the policy model instead of some heuristic data augmentation methods. Alongside adversarial training, the policy model is updated by Algorithm 2 every K iterations of updating the target model. Please let us know if anything is still unclear.
>
> > Table 2 reports the results of the proposed AROID method combined with different AT methods. But it is not described how they are combined.
> >
>
> AROID is combined with other AT methods in the same way as any other data augmentation: simply use the sampled data augmentation policy to augment the data before generating adversarial examples. The update of the policy model is independent of the adversarial training method used. This information has been added to our paper in appendix G.2.
>
> > The results on ImageNet are not convincing enough.It seems the proposed method works well on small datasets. But the results on the large dataset ImageNet are not convincing enough. In ref [a], it is reported that ConvNeXt-T with random init., basic augment. + strong clean pre-training + heavy augmentations achieved 46.5% robust accuracy (see Table 1 in [a]). But this work only achieved 40.40% robust accuracy. It seems that the proposed data augmentation method is not as effective as the method in [a]. Anyway, I think it is necessary to compare with the method in [a]. About the ImageNet experiment, if the experimental settings in [a] are used, e.g., run experiments upon (random init., basic augment. + strong clean pre-training), can AROID show advantage over heavy augmentations proposed in [a]?
> >
>
> We believe that directly comparing our result against [a] is unfair. The experiment setting of [a] is substantially different from ours and several techniques they used are advantageous. For example, they use a stronger attack APGD compared to our naïve PGD to generate training adversarial examples. They use EMA and label smoothing to enhance the performance. The batch size is 1392 while ours is 128, which has been observed by [1] to benefit generalization on ImageNet. The optimizer and learning rate schedule are AdamW and cosine decay which are different from our SGD and multi-step decay.
>
> Instead of comparing AROID against the heavy data augmentation of [a], we find it may be more useful to apply AROID to optimize it. The heavy data augmentation is composed of several individual data augmentation methods, RandAugment, CutMix, MixUp and Random Erasing, and each of them has a few hyperparameters jointly constituting a large search space. As we learn from [a], they did not sufficiently search this space to optimize the performance. We consider this as an opportunity to apply our method to automatically and efficiently optimize these hyperparameters to improve on the original scheme. Unfortunately, training on ImageNet is extremely computationally expensive for our current resource and we cannot provide the result within the period of rebuttal. However, we are happy to provide these results once done in the future revised manuscript. We have added this discussion to our paper.
>
> [1] Goyal et al., Accurate, large minibatch sgd: Training imagenet in 1 hour, Arxiv 2017

---

> > ### Comment · Reviewer_Dov6 · 2023-11-22
> >
> > Thanks for the clarification. I do think that comparison with ref. [a] or combining the proposed method based on ref. [a] is important to show the advantage of the proposed method. I decide to keep my original rating.

---

> > > ### Author Response · Authors · 2023-11-22
> > >
> > > Thanks for your reply and recognizing our work. We do agree with you on the value of comparing or combining with ref. [a] and will update the relevant discussion in our revised manuscript if any result is available. Please let us know if anything else we can help.

---

### Author Response · Authors · 2023-11-21
**Shared response**

We would like to thank the reviewers for taking the time to review our paper and for all the constructive feedback. The paper has been revised (marked blue) according to the comments. We show below the new result of our method when trained with [extra real data](https://github.com/yaircarmon/semisup-adv) and compare it against the leading robust models on the [CIFAR10 Linf leaderboard](https://robustbench.github.io/index.html#div_cifar10_Linf_heading) from RobustBench. We select BDM [1] as the representative leading robustness method since it is ranked 2nd on RobustBench (the 1st method focus on designing robust architectures which is essentially different from our approach). We also select HAT [2] and PORT [3] to compare because they are ranked 1st and 2nd on RobustBench for WRN-34-10 models which is the architecture we used.

We highlight that

1. AROID achieves a robustness matching HAT’s and close to BDM (400 epochs)’s and a clearly higher accuracy exhibiting a better tradeoff between accuracy and robustness. Note that the result of AROID can be further improved by using their advantegous training set-ups such as more extra data, SiLU variant of WRN-34-10 and more effective AT method.
2. Although our best robustness of 62.60% is behind the leading robustness record of BDM (2000 epochs), 70.69%, on RobustBench, we argue that directly comparing these numbers is unfair given the huge advantage of BDM’s training set-up over ours: BDM uses 100x more extra data, larger model architecture, 5x more training epochs, 2x larger batch size and more effective AT method.
3. Overall, the results confirm **the effectiveness of AROID incorporating with extra data** and reveal the **potential of AROID as a complementary techniques for SOTA methods to enhance the state-of-the-art adversarial robustness**. We emphasize that **AROID is a data augmentation method instead of "extra data" method so it does not have to beat BDM**. The key point is that AROID can be combined with extra data to further improve accuracy and robustness.

| Method | Ranking on RobustBench | Epochs | Batch Size | Extra Data | Model | AT | Accuracy | Robustness |
| --- | --- | --- | --- | --- | --- | --- | --- | --- |
| PORT | #2 of WRN-34-10 models | 200 | 128 | 10M Synthetic | WRN34-10-ReLU | PGD10 | 86.68 | 60.27 |
| HAT | #1 of WRN-34-10 models | 400 | 512 | 0.5M Real | WRN34-10-SiLU | HAT | 91.47 | 62.83 |
| BDM | - | 400 | 512 | 1M Synthetic | WRN28-10-SiLU | TRADES | 91.12 | 63.35 |
| BDM | #2 of all models | 2000 | 1024 | 50M Synthetic | WRN70-16-SiLU | TRADES | 93.25 | 70.69 |
| AROID | - | 200 | 128 | 0.5M Real | WRN34-10-ReLU | PGD10 | 92.38 | 61.49 |
| AROID | - | 400 | 512 | 0.5M Real | WRN34-10-ReLU | PGD10 | 92.57 | 62.60 |

Reference:

[1] Wang et al., Better Diffusion Models Further Improve Adversarial Training, ICML 2023

[2] Rade et al., Reducing Excessive Margin to Achieve a Better Accuracy vs. Robustness Trade-off, ICLR 2022

[3] Sehwag et al., Robust Learning Meets Generative Models: Can Proxy Distributions Improve Adversarial Robustness?, ICLR 2022